# Learning Rate Annealing Improves
# Tuning Robustness in Stochastic Optimization

**Amit Attia** [1]   **Tomer Koren** [1,2]

## Abstract

The learning rate in stochastic gradient methods is a critical hyperparameter that is notoriously costly to tune via standard grid search, especially for training modern large-scale models with billions of parameters. We identify a theoretical advantage of learning rate annealing schemes that decay the learning rate to zero at a polynomial rate, such as the widely-used cosine schedule, by demonstrating their increased robustness to initial parameter misspecification due to a coarse grid search. We present an analysis in a stochastic convex optimization setup demonstrating that the convergence rate of stochastic gradient descent with annealed schedules depends *sublinearly* on the multiplicative misspecification factor $\rho$ (i.e., the grid resolution), achieving a rate of $O(\rho^{1/(2p+1)}/\sqrt{T})$ where $p$ is the degree of polynomial decay and $T$ is the number of steps. This is in contrast to the $O(\rho/\sqrt{T})$ rate obtained under the inverse-square-root and fixed stepsize schedules, which depend linearly on $\rho$. Experiments confirm the increased robustness compared to tuning with a fixed stepsize, that has significant implications for the computational overhead of hyperparameter search in practical training scenarios.

## 1. Introduction

Stochastic Gradient Descent (SGD, Robbins & Monro, 1951) is a cornerstone of modern machine learning. Starting at a point $x_1$, the update step of SGD takes the form $x_{t+1} = x_t - \eta_t g_t$, where $\eta_t$ is the stepsize at step $t$ and $g_t$ is a stochastic gradient at $x_t$. An effective stepsize sequence $\eta_1, \eta_2, \ldots$ is critical for performance, yet it is notoriously hard to tune in many scenarios and applications (e.g., Bot-

tou, 2012; Schaul et al., 2013). Furthermore, as models continue to scale, the computational burden of stepsize tuning becomes increasingly demanding.

A common approach to tuning the stepsize sequence is simply using a fixed stepsize, selecting the best fixed value by performing a geometric grid search (Bengio, 2012). In this method, the stepsize is selected based on its performance on a validation set, with the grid resolution determining the (multiplicative) proximity to the best stepsize within the specified range.

A primary approach to moving beyond fixed stepsize sequences is stepsize scheduling. In stepsize scheduling (e.g., Smith, 2017; Loshchilov & Hutter, 2017; Ge et al., 2019), the step at time $t$ is determined by multiplying a baseline stepsize parameter with a parametric sequence. While the approach enables more versatile stepsize sequences and often leads to improved performance, it still requires tuning the baseline stepsize parameter, typically through grid search. Some stepsize schedules also exhibit theoretical benefits, such as anytime convergence guarantees and better last-iterate guarantees (e.g., Jain et al., 2019; Zamani & Glineur, 2025; Liu & Zhou, 2024; Defazio et al., 2024a).

While stepsize tuning is a widely adopted practice, its theoretical foundations remain under-explored. One key question is how sensitive this procedure is to the grid resolution. Limited computational budgets restrict the resolution of grid searches, an issue that has become increasingly prominent with the emergence of modern models consisting of billions of parameters that take days—sometimes weeks—to train. In fact, at massive scales, it is often the case that *any* methodological tuning of the stepsize is prohibitive and therefore abandoned entirely.

Standard analyses of SGD in the convex setting demonstrate a linear degradation in convergence rate as a function of the multiplicative misspecification of the stepsize, which can be significant when performing a coarse—or even absent— grid search. This work investigates to what extent stepsize schedules can mitigate this dependency, providing more robust performance at lower grid resolutions.

Focusing our analysis on stochastic *convex* optimization, we establish convergence guarantees for SGD with stepsize

---

[1]Blavatnik School of Computer Science and AI, Tel Aviv University [2]Google Research Tel Aviv. Correspondence to: Amit Attia <amitattia@mail.tau.ac.il>.

*Proceedings of the 43$^{rd}$ International Conference on Machine Learning*, Seoul, South Korea. PMLR 306, 2026. Copyright 2026 by the author(s).

schedules that decay polynomially to zero, which reveals a key advantage of automatically adapting to multiplicative overestimation of the stepsize. For commonly used schedules, such as cosine annealing, our guarantees yield a *sublinear dependence on the misspecification factor*, in contrast to the linear dependence that arises with the inverse-square-root and fixed stepsize schedules. We further validate our theoretical findings through experiments on synthetic and real data, demonstrating improved robustness to stepsize tuning using decaying schedules compared to tuning a constant stepsize using a grid-search.

## 1.1. Summary of contributions

In more detail, we consider stochastic first-order convex optimization settings, where we aim to minimize a convex objective $f : \mathcal{X} \to \mathbb{R}$, where $\mathcal{X} \subset \mathbb{R}^d$ is a convex set with diameter $D$, while accessing $f$ only through a (sub-)gradient oracle $g$ (i.e., $\mathbb{E}[g(x)] \in \partial f(x)$ for all $x \in \mathcal{X}$). Given an initial stepsize $\eta > 0$, it will be convenient for our development to view a stepsize schedule as being specified by a continuous function $h : [0, 1] \to [0, 1]$ through $\eta_t = \eta h(\frac{t-1}{T})$, where $T$ is the total number of SGD update steps.[1]

We make the following contributions:

- Our main result is a convergence guarantee for (the last iterate of) $T$-steps SGD using a decaying schedule $h$, supporting both the convex Lipschitz case and the convex $\beta$-smooth case. The convergence guarantee is of the form

$$O\left(\mathsf{Rate}_{h,T}^{\mathsf{tu}}\right) \cdot \inf_{\tau \in [\tau_0, 1)} \left\{ \frac{1}{\rho H_h(\tau)} + \rho Q_h(\tau) \right\},$$

  where $\mathsf{Rate}_{h,T}^{\mathsf{tu}}$ is the convergence rate using a *tuned* stepsize, $\rho = \eta/\eta_{\mathsf{tu}} \geq 1$ is the multiplicative overestimation factor of $\eta$ compared to the tuned stepsize $\eta_{\mathsf{tu}}$, and $H_h$ and $Q_h$ are certain functions that depend only on the schedule $h$ and are defined in Equation (1). In the Lipschitz case we may take $\tau_0$ to be zero, and in the smooth case it is the fraction of steps with $\eta_t > 1/2\beta$.[2] When $\tau_0$ is sufficiently small (or zero), the infimum above is at most $O(\rho)$ for any schedule, but, as we show next through key applications of the main result, it can become sublinear in $\rho$ for certain annealed schedules.

- Applying our main result to the cosine annealing schedule in the convex Lipschitz case, we obtain that the last-iterate convergence rate of SGD is $O(\rho^{0.2}DG/\sqrt{T})$. Similarly,

---

[1]While simple discrete schedules can be handled directly, the analysis of more complex schedules will be more conveniently done through integrations of the continuous representation rather than with discrete sums.

[2]The dependence on $\tau_0$ in the smooth case is unavoidable (up to constants), as convergence in smooth optimization requires step size smaller than $2/\beta$.

applying the same result to the polynomially decaying schedule $(1 - \frac{t-1}{T})^p$ for a constant degree $p \geq 1$, we obtain that the last-iterate convergence rate of SGD is $O(\rho^{1/(2p+1)}DG/\sqrt{T})$. In the convex smooth case, assuming $\eta_1 = \eta h(0) \leq 1/2\beta$, we obtain the same multiplicative sub-optimality of $\rho^{0.2}$ and $\rho^{1/(2p+1)}$ for cosine annealing and (degree $p$-)polynomially decaying schedules.

- Additionally, we validate the robustness of various learning rate schedules to tuning in experiments, by performing grid search on two tasks: a synthetic logistic regression task with a linear model and the CIFAR-10 classification task with a deep neural network. We find that, when using a coarse grid, annealing schemes—specifically cosine annealing and linear decay—demonstrate greater robustness compared to a fixed step size schedule.

Our theoretical results show that polynomially decaying schedules, including cosine annealing, achieve convergence rates with a sublinear dependence on the misspecification factor, in contrast to the linear dependence observed in SGD with the inverse-square-root schedule ($\eta_t = \eta/\sqrt{t}$) and the fixed stepsize schedule–a dependence that we show to be unavoidable in Section C. This distinction is particularly striking since, while both fixed and annealed stepsizes are able to attain the optimal convergence rate when properly tuned, the latter exhibits significantly greater robustness to parameter misspecification. When tuning the stepsize using a coarse grid search under a limited computational budget, this difference in robustness can significantly impact performance, as also illustrated in our synthetic and real-data experiments.

Due to space constraints, we focus below on the convex Lipschitz case and defer the treatment of the convex smooth case to Section A.

## 1.2. Overview of approach and key ideas

Before describing the key ideas of our approach, it is illustrative to first consider a straightforward strategy for bounding the convergence of fixed-stepsize SGD when the stepsize is overestimated by a factor $\rho > 1$ relative to the tuned stepsize. For convex Lipschitz objectives, the standard average-iterate convergence rate of fixed-stepsize SGD is $O(D^2/(\eta T) + \eta G^2)$, where $G^2$ denotes a bound on the second-moment of the stochastic gradients. Plugging in an overestimated stepsize of $\eta = \rho D/(G\sqrt{T})$, which is larger than the worst-case optimal stepsize by a factor of $\rho$, we obtain a bound that degrades *linearly* in $\rho$ due to the linear dependence on $\eta$ in the second term.

While this linear degradation in $\rho$ is unavoidable for fixed stepsize SGD (see Section C), our key insight is that pairing it with an annealing schedule reduces the misspecification at some (unknown) point in time, due to gradual stepsize

decay. To exploit this property in the analysis, we avoid "unrolling" the standard convergence analysis all the way from the final iterate $x_{T+1}$ to the initial point $x_1$; instead, we analyze the algorithm starting from some intermediate iterate $x_k$, where the stepsize $\eta_k$ has sufficiently decayed so as to mitigate the initial misspecification.

A challenge with this approach lies in the algorithm's output: standard analyses of SGD typically rely on (possibly weighted) averages of the iterates, but since the optimal starting index $k$ is unknown in advance (as it depends on unknown problem parameters), we cannot determine a suitable averaging scheme. To circumvent this, we turn from an average-iterate to a last-iterate analysis of SGD with varying stepsizes, which avoids this issue as the final point is independent of the choice of $k$. By combining these observations, we can analyze the convergence of SGD from an implicitly specified "optimal" starting point $k$ and show that it yields a strictly improved robustness to the initial stepsize misspecification, despite the smaller number of steps that remained for convergence after step $k$.

### 1.3. Related work

**Adaptive and parameter-free methods.** Beyond learning rate scheduling, several approaches have been developed to minimize the need for extensive tuning in first-order optimization. These include adaptive methods, such as AdaGrad and Adam (e.g., Duchi et al., 2011; Kingma & Ba, 2015), as well as recent theoretical advancements (Reddi et al., 2018; Tran et al., 2019; Kavis et al., 2019; Alacaoglu et al., 2020; Faw et al., 2022; Kavis et al., 2022; Attia & Koren, 2023; Liu et al., 2023), which utilize gradient statistics to dynamically adjust learning rates. Additionally, parameter-free methods (e.g., Chaudhuri et al., 2009; Streeter & McMahan, 2012; Luo & Schapire, 2015; Orabona & Pál, 2016; Cutkosky & Orabona, 2018; Orabona & Pál, 2021; Carmon & Hinder, 2022) primarily focus on automatically adapting to the problem's complexity, such as the distance to the optimal solution. Recently, several parameter-free approaches demonstrated impressive practical performance, narrowing the gap to finely-tuned methods (Ivgi et al., 2023; Defazio & Mishchenko, 2023; Mishchenko & Defazio, 2024). While these approaches take different paths to reduce tuning, adaptive methods and scheduling schemes are often used together in practice.

**Theoretical analyses of stepsize annealing.** Several studies have analyzed different stepsize schedules. The influential work of Jain et al. (2019) showed that the schedules $\eta_t = \eta/t$ and $\eta_t = \eta/\sqrt{t}$ yield suboptimal last-iterate guarantees and proposed a new schedule with optimal last-iterate performance. Later, Defazio et al. (2024a) demonstrated that a linear decay schedule also achieves an optimal last-iterate guarantee. Additionally, Defazio et al. (2024b) introduced

"schedule-free" SGD, which eliminates the need to know the training length $T$ in advance. While these works focus on optimality with well-tuned stepsizes and last-iterate guarantees, our work examines the robustness of these schedules when the step size is not finely tuned. Under additional assumptions, different stepsize schedules demonstrated other benefits compared to the fixed stepsize schedule, including improved convergence rate for quadratic objectives (Goujaud et al., 2022) and improved noise adaptivity under the Polyak-Łojasiewicz condition (Li et al., 2021), and Wu et al. (2022) compared different stepsize decaying schedules in the context of overparameterized linear regression. Additionally, new scheduling schemes continue to emerge, such as those proposed by Zhai et al. (2022) and Hu et al. (2024), which incorporate a cooldown phase to accommodate varying training durations. The robustness perspective we propose helps us better understand the benefits of different schedules and guides the design of more robust ones.

## 2. Preliminaries

### 2.1. Problem setup

In this work, we are interested in first-order stochastic optimization over a bounded domain within the $d$-dimensional Euclidean space, $\mathbb{R}^d$, equipped with the Euclidean norm, defined as $\|\cdot\| \triangleq \|\cdot\|_2$. Let $\mathcal{X} \subset \mathbb{R}^d$ be a convex set with diameter $D$ (i.e., for all $x, y \in \mathcal{X}$, $\|x - y\| \leq D$) and let $f : \mathcal{X} \rightarrow \mathbb{R}$ be a convex function. Our goal is to find some $\overline{x} \in \mathcal{X}$ such that $f(\overline{x}) - \min_{x \in \mathcal{X}} f(x)$ is small, where we access $f$ only through an unbiased sub-gradient oracle $g : \mathcal{X} \rightarrow \mathbb{R}^d$ (i.e., $\mathbb{E}[g(x)] \in \partial f(x)$ for all $x \in \mathcal{X}$, where we denote with a slight abuse of notation $\nabla f(x) \triangleq \mathbb{E}[g(x)]$). We consider two optimization scenarios:

(i) ***Convex and Lipschitz setting.*** Here we assume $g$ has a second moment bound, that is, for some $G > 0$, $\mathbb{E}\|g(x)\|^2 \leq G^2$ for all $x \in \mathcal{X}$. This implies in particular that $f$ is $G$-Lipschitz.

(ii) ***Convex and smooth setting.*** In this scenario we assume that $f$ is $\beta$-smooth,[3] and instead of a second moment bound we assume that $g$ has a variance bound, that is, for some $\sigma > 0$, $\mathbb{E}[\|g(x) - \nabla f(x)\|^2] \leq \sigma^2$ for all $x \in \mathcal{X}$. Due to space constraints, the treatment of this case is deferred to Section A.

**Stochastic Gradient Descent (SGD).** We will analyze the (projected) SGD algorithm, which starts at some $x_1 \in \mathcal{X}$ and performs update steps of the form $x_{t+1} = \Pi_{\mathcal{X}}(x_t - \eta_t g_t)$, where $\eta_t$ is the stepsize at step $t$, $g_t = g(x_t)$ is a stochastic

---

[3]A function $f : \mathcal{X} \rightarrow \mathbb{R}$ is said to be $\beta$-smooth if $\|\nabla f(x) - \nabla f(y)\| \leq \beta\|x - y\|$ for all $x, y \in \mathcal{X}$. In particular, this implies that $|f(y) - f(x) - \nabla f(x) \cdot (y - x)| \leq \frac{\beta}{2}\|y - x\|^2$ for all $x, y \in \mathcal{X}$.

sub-gradient at $x_t$, and $\Pi_{\mathcal{X}}(\cdot)$ is the Euclidean projection to $\mathcal{X}$. The output of $T$-steps SGD is typically some average of the iterates or the last iterate. The convergence rate guarantee of the average iterate of fixed stepsize SGD with tuned stepsize is $O(DG/\sqrt{T})$ in the convex Lipschitz case and $O(\beta D^2/T + D\sigma/\sqrt{T})$ in the convex smooth case (See, e.g., Lan, 2012).

**Stepsize scheduling.** Our focus will be on stepsizes of the form $\eta_t = \eta h(\frac{t-1}{T})$, for some $\eta > 0$ and $h : [0, 1] \to [0, 1]$, where $T \in \mathbb{N}$ is the number of SGD steps. Common schedules include $h(u) = 1$ (fixed stepsize), $h(u) = \frac{1}{2} + \frac{1}{2}\cos(\pi u)$ (cosine annealing), and $h(u) = (1 - u)^p$ for some $p \geq 1$ (polynomial decay). In particular, we will assume that $h(u)$ is monotonically non-increasing, and satisfy $h(u) = 0 \Leftrightarrow u = 1$; we will call such a schedule *annealed* for brevity. We additionally assume for technical reasons that the annealed schedules we consider are differentiable and $p$-Lipschitz. Using an annealed schedule, SGD with a properly tuned step size yields the same rate as optimally tuned fixed step-size SGD, up to constant factors (where we treat $p$ as a constant). See Section D for additional details. Notable annealed schedules include cosine annealing and polynomial decay.

**Robustness to stepsize misspecification.** Fixing an initialization $x_1 \in \mathcal{X}$ and a stepsize schedule $h(\cdot)$, it remains to tune the base stepsize $\eta$. Considering a tuned stepsize $\eta_{\mathsf{tu}}$,[4] we investigate the sensitivity of SGD when the stepsize is only tuned to a multiplicative misspecification factor $\rho \geq 1$ (i.e., stepsize $\eta = \rho\eta_{\mathsf{tu}}$, where $\rho$ is of course unknown to the algorithm).[5] In this case, the convergence rate will likely degrade as $\rho$ increases. For instance, the standard guarantee of fixed stepsize SGD degrades linearly in $\rho$; we demonstrate this fact in the convex Lipschitz setting in Section C.

Our main inquiry is to what extent stepsize schedules can mitigate this degradation, enabling more robust performance when the stepsize is crudely tuned (e.g., when tuned using a coarse grid search, where the overshoot factor can be large enough to have a non-negligible effect on the convergence rate), and achieving convergence rates with sublinear dependence on $\rho$, for $\rho \geq 1$.

We remark that adaptive and parameter-free methods provide other avenues for robustness beyond stepsize scheduling, which are beyond the scope of this work; see Section 1.3 for details.

---

[4]By "tuned" we mean a stepsize that minimize a corresponding convergence guarantee that depend on $\eta$, possibly ignoring lower-order terms for simplicity.

[5]We focus on $\rho \geq 1$ because (A) grid search necessarily includes values that overshoot the optimal learning rate, and (B) in the context of decaying schedules, annealing is expected to be beneficial when the learning rate is overestimated.

## 2.2. Convergence analysis with stepsize schedules

Here we present a convergence guarantee for SGD using an annealed schedule. The tuned stepsize and respective convergence rate will serve as the baseline for establishing a sublinear dependence on the misspecification parameter. For the proof, see Section D.

Let $h$ be a differentiable $p$-Lipschitz annealed schedule $h$. We define the following two functions associated with $h$:

$$H_h(v) \triangleq \int_v^1 h(u)\,du, \quad \text{and} \quad Q_h(v) \triangleq \int_v^1 \frac{H_h'(u)^2}{H_h(u)}\,du. \quad (1)$$

Throughout, convergence bounds will be expressed in terms of $H_h$ and $Q_h$. These quantities are continuous analogues of the stepsize-related sums that appear in last-iterate guarantees (e.g., see Section B): for $\eta_t = \eta h((t-1)/T)$ and the suffix $\eta_{k+1}, \ldots, \eta_T$, $\eta T H_h(k/T) \approx \sum_{t=k+1}^T \eta_t$ and $\eta Q_h(k/T) \approx \sum_{t=k+1}^T (\eta_t^2/\sum_{s=t}^T \eta_s)$. We begin with the convex Lipschitz case.

**Lemma 1.** *Let $\mathcal{X} \subset \mathbb{R}^d$ be a convex set with diameter $D > 0$, $f : \mathcal{X} \to \mathbb{R}$ a convex function, $x^\star \in \arg\min_{x \in \mathcal{X}} f(x)$, and $g : \mathcal{X} \to \mathbb{R}^d$ an unbiased first-order oracle of $f$ with second-moment bounded by $G^2 > 0$. Let $x_1, x_2, \ldots, x_{T+1}$ be the iterates produced by $T$-steps SGD with stepsizes $\eta_t = \eta h(\frac{t-1}{T})$ using the oracle $g$, where $h$ is a differentiable $p$-Lipschitz annealed schedule. Then it holds that*

$$\mathbb{E}[f(x_{T+1}) - f(x^\star)] \leq \frac{D^2}{2\eta T H_h(0)} + 2\eta G^2 Q_h(0) + \frac{8p\eta G^2}{T}.$$

We denote the tuned stepsize and respective convergence guarantee (up to lower-order terms) by

$$\begin{aligned} \eta_{\mathsf{tu}} &\triangleq \frac{D}{2G\sqrt{T H_h(0) Q_h(0)}}; \\ \mathsf{Rate}_{h,T}^{\mathsf{tu}} &\triangleq \frac{2DG}{\sqrt{T}}\sqrt{Q_h(0)/H_h(0)}. \end{aligned} \quad (2)$$

As previously mentioned, under the mild assumption $p = \Theta(1)$ (the Lipschitz parameter of $h$), the guarantees match the rates of optimally tuned fixed stepsize SGD (see Section D for details).

**Remark.** A straightforward baseline approach to account for stepsize overestimation is to substitute $\eta = \rho\eta_{\mathsf{tu}}$ for misspecification factor $\rho > 1$ into Lemma 1. Under this substitution, the second term in the bound of Lemma 1 becomes

$$2\eta G^2 Q_h(0) = \rho DG\sqrt{Q_h(0)/H_h(0)} = \frac{\rho}{2}\mathsf{Rate}_{h,T}^{\mathsf{tu}},$$

implying that the bound deteriorates at least linearly with $\rho$ (neglecting lower-order terms). Our goal in this work is to improve over such a linear deterioration and obtain bounds that scale sublinearly with $\rho$.

## 3. Convex and Lipschitz setting

This section considers a convex objective where the second moment of the sub-gradient oracle is bounded. The main result of this section is a convergence guarantee that mitigates the imbalance caused by overestimation by automatically adapting to the tails of $H_h(v)$ and $Q_h(v)$. The key observation in obtaining this result is that any suffix of iterates $x_k, \dots, x_{T+1}$ can be viewed as a $(T-k+1)$-steps SGD starting at $x_k$, effectively ignoring the large stepsizes prior to step $k$ that would otherwise degrade the convergence bound.

Next, we present the general guarantee, followed by corollaries for specific schedules.

**Theorem 1.** *Let $X \subset \mathbb{R}^d$ be a convex set with diameter $D > 0$, $f : X \to \mathbb{R}$ a convex function, $x^\star \in \arg\min_{x \in X} f(x)$, and $g : X \to \mathbb{R}^d$ an unbiased first-order oracle of $f$ with second-moment bounded by $G^2 > 0$. For any $\rho \geq 1$, let $x_1, x_2, \dots, x_{T+1}$ be the iterates produced by $T$-steps SGD with stepsizes $\eta_t = \eta h(\frac{t-1}{T})$ using the oracle $g$, where $\eta = \rho \cdot \eta_{\text{tu}}$ and $h$ is a differentiable $p$-Lipschitz annealed schedule. Then it holds that*

$$\mathbb{E}[f(x_{T+1}) - f(x^\star)] \leq \tfrac{1}{2} \mathsf{Rate}_{h,T}^{\text{tu}} \cdot \inf_{\tau \in [0,1)} \left( \frac{H_h(0)}{\rho H_h(\tau)} + \frac{\rho Q_h(\tau)}{Q_h(0)} \right)$$
$$+ O\left( \frac{p\rho\eta_{\text{tu}}G^2}{T} \right), \tag{3}$$

*where $H_h$, $Q_h$, $\eta_{\text{tu}}$, and $\mathsf{Rate}_{h,T}^{\text{tu}}$ are given in Equations (1) and (2). In particular, the optimal $\tau$ satisfies $H_h(\tau)H_h'(\tau) = -H_h(0)Q_h(0)/\rho^2$ (or $\tau = 0$ if there is no solution).*

First, note that for $\rho = 1$, Theorem 1 recovers $\mathsf{Rate}_{h,T}^{\text{tu}}$ up to low order terms, as the infimum is at most 2. Furthermore, as $\rho \geq 1$ and both $H_h(v)$ and $Q_h(v)$ are decreasing and equal 0 at $v = 1$, the infimum adapts to the imbalance of the $\frac{1}{\rho}$ and $\rho$ terms which are introduced by the overestimation.

We defer the proof of Theorem 1 to Section 3.1. Following are corollaries for polynomially decaying and cosine annealing schedules which provide concrete examples for the power of Theorem 1.

**Corollary 2.** *In the setting of Theorem 1, assuming $h(u) = (1-u)^p$ for some $p \geq 1$,*

$$\mathbb{E}[f(x_{T+1}) - f(x^\star)] = \mathsf{Rate}_{h,T}^{\text{tu}} \cdot \rho^{\frac{1}{2p+1}} + O\left( \frac{p\rho\eta_{\text{tu}}G^2}{T} \right),$$

*where $\mathsf{Rate}_{h,T}^{\text{tu}} = \frac{2(p+1)}{\sqrt{p}} \cdot \frac{DG}{\sqrt{T}} = O\left( \frac{\sqrt{p}DG}{\sqrt{T}} \right)$.*

We observe that for $p = \Theta(1)$, the optimal rate is the same as tuned SGD with fixed stepsize (up to constants), while the dependence on $\rho \geq 1$ is sublinear, as we aimed to achieve. The dependence $\rho^{\frac{1}{2p+1}}$ might lead to the idea that a larger $p$ is always better, but as $p$ increases the optimal rate degrades

at a rate of $O(\sqrt{p})$. In particular, using $p = \Theta(\log \rho)$ the convergence rate will be $O(DG\sqrt{\log \rho}/\sqrt{T})$, and increasing beyond this point will not improve the final rate.

*Proof of Corollary 2.* First note that $h(u) = (1-u)^p$ is non-increasing, differentiable, $p$-Lipschitz (since $|h'(u)| = p(1-u)^{p-1} \leq p$) and satisfies $h(u) = 0 \Leftrightarrow u = 1$. Hence, $h$ is *annealed* and we can use Theorem 1. Integrating, $H_h(\tau) = \frac{1}{p+1}(1-\tau)^{p+1}$, and $H_h'(\tau) = -(1-\tau)^p$. Thus,

$$Q_h(\tau) = \int_\tau^1 \frac{H_h'(u)^2}{H_h(u)} du = \frac{p+1}{p}(1-\tau)^p.$$

We proceed to solve the optimality equation of Theorem 1, $H_h(\bar{\tau})H_h'(\bar{\tau}) = -H_h(0)Q_h(0)/\rho^2$:

$$\frac{-(1-\bar{\tau})^{2p+1}}{p+1} = \frac{-1}{p\rho^2} \implies \bar{\tau} = 1 - \left( \frac{p+1}{p\rho^2} \right)^{\frac{1}{2p+1}}.$$

While this value is optimal, it may be negative for small $\rho$, so we select a slightly sub-optimal value of $\bar{\tau} = 1 - \rho^{\frac{-2}{2p+1}} \in [0,1)$ which is always valid. Using this value,

$$\frac{H_h(0)}{\rho H_h(\bar{\tau})} + \frac{\rho Q_h(\bar{\tau})}{Q_h(0)} = \frac{(1-\bar{\tau})^{-(p+1)}}{\rho} + \rho(1-\bar{\tau})^p$$
$$= \frac{\rho^{\frac{2(p+1)}{2p+1}}}{\rho} + \rho\rho^{\frac{-2p}{2p+1}} = 2\rho^{\frac{1}{2p+1}}. \tag{4}$$

Hence, using this value to bound the infimum of Equation (3),

$$\mathbb{E}[f(x_{T+1}) - f(x^\star)] \leq \mathsf{Rate}_{h,T}^{\text{tu}} \cdot \rho^{\frac{1}{2p+1}} + O\left( \frac{p\rho\eta_{\text{tu}}G^2}{T} \right).$$

We conclude by plugging the values of $H_h(0)$ and $Q_h(0)$ to Equation (2) (and using $p \geq 1$). □

We proceed to the cosine annealing guarantee. The proof is deferred to Section G.

**Corollary 3.** *In the setting of Theorem 1, assuming $h(u) = \frac{1}{2}(1 + \cos(\pi u))$,*

$$\mathbb{E}[f(x_{T+1}) - f(x^\star)] \leq \mathsf{Rate}_{h,T}^{\text{tu}} \cdot 18\rho^{\frac{1}{5}} + O\left( \frac{\rho\eta_{\text{tu}}G^2}{T} \right),$$

*where $\mathsf{Rate}_{h,T}^{\text{tu}} = \frac{2DG}{\sqrt{T}}\sqrt{2Q_h(0)} \leq \frac{10DG}{\sqrt{T}}$.*

Again we observe a sublinear dependence on $\rho$ with an optimal rate of $O(\frac{DG}{\sqrt{T}})$. Note that this is the same behavior as in Corollary 2 with $p = 2$, which arises from the tail behavior of $h(u)$. To see that, one can verify that $(1-u)^2 \leq h(u) \leq \frac{5}{2}(1-u)^2$ for all $u \in [0,1]$. We also remark that the numerical constants in Corollary 3 are not tight; in Section F, we show that improved bounds can be obtained by numerically evaluating the bound in Theorem 1 for the cosine annealing schedule.

### 3.1. Proof of Theorem 1

Before proving our main theorem, we first state a few lemmas we will require. The first is a last-iterate convergence guarantee, using the techniques of Zamani & Glineur (2025); Liu & Zhou (2024) (proof appearing in Section B).

**Lemma 2.** *Let $\mathcal{X} \subseteq \mathbb{R}^d$ be a convex set, $f : \mathcal{X} \to \mathbb{R}$ a convex function, and $g : \mathcal{X} \to \mathbb{R}^d$ an unbiased first-order oracle of $f$ with second-moment bounded by $G^2 > 0$. Let $x_1, \ldots, x_{T+1}$ be the iterates produced by $T$-steps SGD with stepsizes $\eta_1, \ldots, \eta_T$ using the oracle $g$. Then for any $\hat{x} \in \mathcal{X}$,*

$$\mathbb{E}[f(x_{T+1}) - f(\hat{x})] \leq \frac{\|x_1 - \hat{x}\|^2}{2\sum_{s=1}^T \eta_s} + 2G^2 \sum_{t=1}^T \frac{\eta_t^2}{\sum_{s=t}^T \eta_s}.$$

Next is a key lemma, translating the suffix of the last-iterate bound in Lemma 2 to one based on integrating the stepsize schedule (proof given later in the section).

**Lemma 3.** *Let $k \in [T]$, $c_1, c_2, \eta > 0$, and $\eta_t = \eta h(\frac{t-1}{T})$ for some differentiable $p$-Lipschitz annealed schedule $h$. Then for any $\tau \in [\frac{k-1}{T}, \frac{k}{T})$,*

$$\frac{c_1}{\sum_{s=k}^T \eta_s} + c_2 \sum_{t=k}^T \frac{\eta_t^2}{\sum_{s=t}^T \eta_s}$$

$$\leq \frac{c_1}{\eta T H_h(\tau)} + c_2 \eta \int_\tau^{1 - \frac{1}{T}} \frac{h(u)^2}{H_h(u)} du + \frac{4\eta c_2 p}{T}.$$

We proceed to the proof of Theorem 1.

*Proof of Theorem 1.* Let $\tau \in [0, 1)$ and let $k = \lfloor \tau T \rfloor + 1 \in [T]$. Consider the suffix $x_k, x_{k+1}, \ldots, x_{T+1}$ as an SGD sequence starting at $x_k$. By Lemma 2 with $\hat{x} = x^\star$,

$$\mathbb{E}[f(x_{T+1}) - f(x^\star)] \leq \frac{D^2}{2\sum_{s=k}^T \eta_s} + 2G^2 \sum_{t=k}^T \frac{\eta_t^2}{\sum_{s=t}^T \eta_s}.$$

As $\tau \in [\frac{k-1}{T}, \frac{k}{T})$, by Lemma 3 with $c_1 = \frac{D^2}{2}$ and $c_2 = 2G^2$,

$$\mathbb{E}[f(x_{T+1}) - f(x^\star)]$$
$$\leq \frac{D^2}{2\eta T H_h(\tau)} + 2\eta G^2 \int_\tau^1 \frac{h(u)^2}{H_h(u)} du + \frac{8p\eta G^2}{T}$$
$$= \frac{1}{2}\mathsf{Rate}_{h,T}^{\mathrm{tu}} \cdot \left( \frac{H_h(0)}{\rho H_h(\tau)} + \frac{\rho \int_\tau^1 \frac{h(u)^2}{H_h(u)} du}{\int_0^1 \frac{h(u)^2}{H_h(u)} du} \right) + \frac{8p\rho\eta_{\mathrm{tu}}G^2}{T}$$
$$\qquad\qquad (\eta = \rho\eta_{\mathrm{tu}} \text{ and Equation (2)})$$
$$= \frac{1}{2}\mathsf{Rate}_{h,T}^{\mathrm{tu}} \cdot \left( \frac{H_h(0)}{\rho H_h(\tau)} + \frac{\rho Q_h(\tau)}{Q_h(0)} \right) + \frac{8p\rho\eta_{\mathrm{tu}}G^2}{T}.$$
$$\qquad\qquad (H_h'(u)^2 = h(u)^2, \text{ Eq. 1})$$

This inequality holds for any $\tau \in [0, 1)$, hence it holds for the infimum over all $\tau \in [0, 1)$. It is left to find the $\tau$ which minimizes the bound. Let

$$g(v) = \frac{H_h(0)}{\rho H_h(v)} + \frac{\rho Q_h(v)}{Q_h(0)}. \tag{5}$$

By the fundamental theorem of calculus, $H_h'(v) = -h(u)$ and

$$Q_h'(v) = \left( \int_0^1 \frac{H_h'(u)^2}{H_h(u)} du - \int_0^v \frac{H_h'(u)^2}{H_h(u)} du \right)' = -\frac{H_h'(v)^2}{H_h(v)},$$

Thus,

$$\begin{aligned} g'(v) &= \frac{-H_h(0)H_h'(v)}{\rho H_h(v)^2} - \frac{\rho \frac{H_h'(v)^2}{H_h(v)}}{Q_h(0)} \\ &= \frac{-\rho H_h'(v)}{Q_h(0)H_h(v)^2} \left( \frac{H_h(0)Q_h(0)}{\rho^2} + H_h(v)H_h'(v) \right) \\ &= \frac{\rho h(v)}{Q_h(0)H_h(v)^2} \left( H_h(v)H_h'(v) + \frac{H_h(0)Q_h(0)}{\rho^2} \right). \end{aligned}$$

Hence, when $\tau$ satisfy $H_h(\tau)H_h'(\tau) = \frac{-H_h(0)Q_h(0)}{\rho^2}$, $g'(\tau) = 0$. For $v > \tau$,

$$\begin{aligned} H_h(v)H_h'(v) &= -H_h(v)h(v) \geq -H_h(v)h(\tau) \\ &> -H_h(\tau)h(\tau) = -\frac{H_h(0)Q_h(0)}{\rho^2}, \end{aligned}$$

so $g'(v) > 0$ and $g(v) > g(\tau)$. Similarly, for $v < \tau$, $g'(v) < 0$ and $g(v) > g(\tau)$. Hence, $\tau$ satisfying $H_h(\tau)H_h'(\tau) = \frac{-H_h(0)Q_h(0)}{\rho^2}$ is the minimizer. If no such $\tau$ exists, the derivative is always positive (as $h$ is continuous and $H_h(1)H_h'(1) = 0$), and the minimizer is at $\tau = 0$. $\qquad\square$

### 3.2. Proof of Lemma 3

Let $\tau \in [\frac{k-1}{T}, \frac{k}{T})$. As $h$ is non-increasing, we can use integration to obtain the following bound,

$$\frac{c_1}{\sum_{t=k}^T \eta_t} + c_2 \sum_{t=k}^T \frac{\eta_t^2}{\sum_{s=t}^T \eta_s}$$

$$\leq \frac{c_1}{\eta \int_k^{T+1} h\left(\frac{t-1}{T}\right) dt} + \frac{c_2}{\eta} \sum_{t=k}^T \frac{\eta_t^2}{\int_t^{T+1} h\left(\frac{s-1}{T}\right) ds}$$

$$= \frac{c_1}{\eta T \int_{\frac{k-1}{T}}^1 h(u) du} + \frac{c_2}{\eta T} \sum_{t=k}^T \frac{\eta_t^2}{\int_{\frac{t-1}{T}}^1 h(u) du}$$

$$= \frac{c_1}{\eta T H_h\left(\frac{k-1}{T}\right)} + \frac{c_2}{\eta T} \sum_{t=k}^T \frac{\eta_t^2}{H_h\left(\frac{t-1}{T}\right)},$$

where the last two inequalities follows by changing integration variables and Equation (1). Again bounding by

integration and changing variables,

$$\frac{c_2}{\eta T}\sum_{t=k}^{T}\frac{\eta_t^2}{H_h\left(\frac{t-1}{T}\right)} \leq \frac{c_2\eta}{T}\left(\frac{h\left(\frac{k-1}{T}\right)^2}{H_h\left(\frac{k-1}{T}\right)} + \int_k^T \frac{h\left(\frac{t-1}{T}\right)^2}{H_h\left(\frac{t-1}{T}\right)}dt\right)$$

$$= \frac{c_2\eta}{T}\left(\frac{h\left(\frac{k-1}{T}\right)^2}{H_h\left(\frac{k-1}{T}\right)} + T\int_{\frac{k-1}{T}}^{1-\frac{1}{T}}\frac{h(u)^2}{H_h(u)}du\right).$$

As $h(u)$ is differentiable, $p$-Lipschitz, and $h(1)=0$, for any $v \in [0,1)$,

$$2pH_h(v) = 2p\int_v^1 h(u)\,du \geq 2\int_v^1 h(u)(-h(u))'\,du \qquad (6)$$
$$= h(v)^2 - h(1)^2 = h(v)^2.$$

Hence, $\frac{h\left(\frac{k-1}{T}\right)^2}{H_h(\frac{k-1}{T})} \leq 2p$ and since $|\tau - \frac{k-1}{T}| \leq \frac{1}{T}$,

$$\int_{\frac{k-1}{T}}^{1-\frac{1}{T}}\frac{h(u)^2}{H_h(u)}du = \int_\tau^{1-\frac{1}{T}}\frac{h(u)^2}{H_h(u)}du + \int_{\frac{k-1}{T}}^{\tau}\frac{h(u)^2}{H_h(u)}du$$
$$\leq \int_\tau^{1-\frac{1}{T}}\frac{h(u)^2}{H_h(u)}du + \frac{2p}{T}.$$

Plugging back,

$$\frac{c_1}{\sum_{t=k}^{T}\eta_t} + c_2\sum_{t=k}^{T}\frac{\eta_t^2}{\sum_{s=t}^{T}\eta_s}$$
$$\leq \frac{c_1}{\eta T H_h(\tau)} + \eta c_2 \int_\tau^{1-\frac{1}{T}}\frac{h(u)^2}{H_h(u)}du + \frac{4\eta c_2 p}{T}. \qquad \square$$

## 4. Experimental evaluation

Our theory predicts that annealing schemes are more robust to learning rate tuning than fixed learning rates. To support the prediction, we perform experiments to compare the performances of different scheduling strategies under varying grid search resolutions for learning rate tuning.

We conduct two types of experiments: the first involves a synthetic logistic regression task closely aligned with the theoretical setting, while the second involves training a neural network classifier.

### 4.1. Experimental setup

We consider common schedules, namely, fixed learning rate (as our baseline), in addition to the decaying cosine annealing, linear decay, and quadratic decay (i.e., $h(u) = (1-u)^2$) schedules. To simulate varying grid resolutions, we train the models using a geometric grid of learning rates with a multiplicative factor of approximately $\sqrt[3]{10} \approx 2.15$ (the values $\{1, 2.2, 5\}$ multiplied by $10^i$ with different $i$'s), and consider

the different subsets with resolutions $2.15, 2.15^2, 2.15^3$, etc. For example, with range $[0.01, 5]$ and resolution of $2.15^3$, we find the best model for each of the grids[6] $\{0.01, 0.1, 1\}, \{0.022, 0.22, 2.2\}, \{0.05, 0.5, 5\}$, and report the average test loss/top-1 error across grids.

**Synthetic logistic regression.** In the synthetic experiment, we generate 100,000 samples of dimension 100, drawn from a normal distribution. Labels are assigned by thresholding probabilities determined by a "true weights" vector of size 100, also sampled from a normal distribution. To introduce additional noise, we flip each label with a probability of 0.1. A test set of the same size is generated similarly. We train a linear classifier using binary cross-entropy loss, SGD without momentum, a batch size of 1,000, and a single epoch (updating the scheduler after each step). For the fixed learning rate scheduler, we report both the last iterate and the averaged iterate performances.

**Wide ResNet on CIFAR-10.** We train a Wide ResNet 28-10 model[7] (Zagoruyko & Komodakis, 2016) without dropout on the CIFAR-10 dataset (Krizhevsky, 2009). We train for 200 epochs, using a batch size of 128, Nesterov momentum of 0.9, and weight decay of 0.0005. The scheduler is updated after each epoch. As the last iterate of fixed stepsize SGD underperforms, we use polynomial averaging (Shamir & Zhang, 2013) with parameter $\gamma = 8$, following Ivgi et al. (2023).

### 4.2. Results

The test loss per learning rate appears in Figure 1 (left). For each resolution, Figure 1 (right) illustrates the logistic regression test loss averaged across the best models for each sub-grid. At high resolutions (e.g., grid parameters up to 10), we observe a comparable performance degradation across different schedules (besides fixed stepsize without averaging which underperforms). However, as grid resolution decreases, the gap between the fixed learning rate schedule and the decaying schedules widens. For instance, with a grid factor of approximately 100, the performance of the fixed learning rate (with averaging) decreases by 0.08, whereas cosine annealing, linear decay, and quadratic decay schedules experience smaller drops of 0.01, 0.014, and 0.08, respectively, with similar trends observed for grids with lower resolutions.

Figure 2 (right) shows the CIFAR-10 top-1 test error for each resolution, averaged over the best models per sub-grid, with the raw test error per learning rate appearing in Figure 2 (left). Similar to the logistic regression task, degradation

---

[6]We average the performance over 3 runs per learning rate.
[7]We use the PyTorch implementation of Wide ResNet at https://github.com/bmsookim/wide-resnet.pytorch.

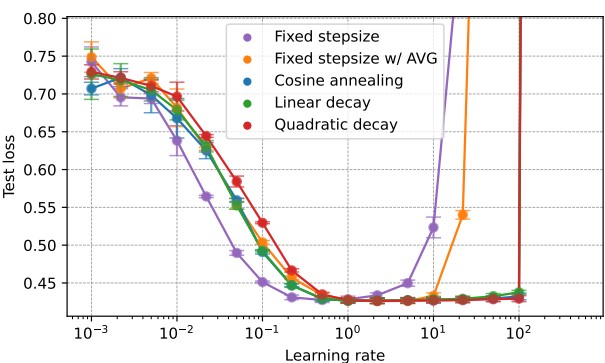 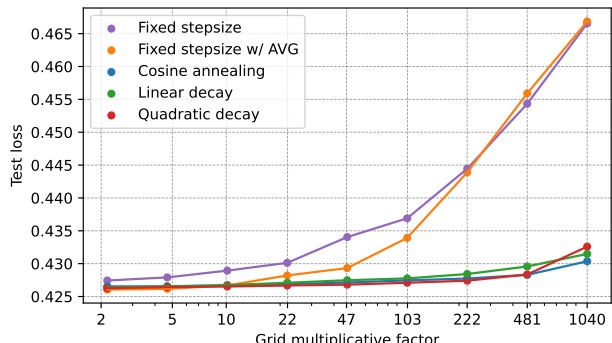

*Figure 1.* (left) Test loss for the synthetic logistic regression task with varying learning rates and different learning rate schedules. Each point represents 3 runs, reporting average and standard deviation. (right) Test loss of the best model in a sub-grid averaged over multiple sub-grids with the same multiplicative grid factor. "Fixed stepsize w/ AVG" stands for fixed stepsize SGD with iterate averaging.

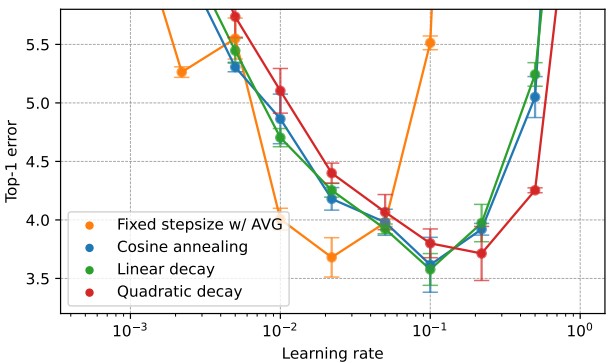 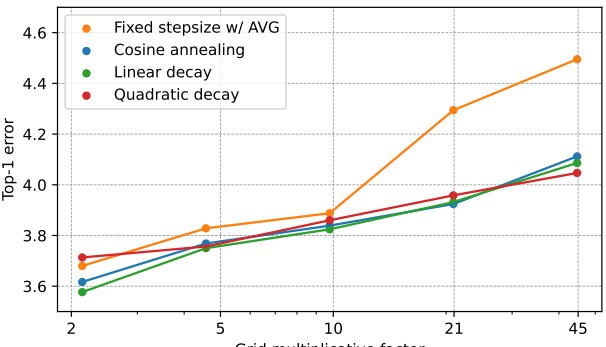

*Figure 2.* (left) CIFAR-10 top-1 test error of WideResNet28-10 with varying learning rates and different learning rate schedules. Each point represents 3 runs, reporting average and standard deviation. (right) Test error of the best model in a sub-grid averaged over multiple sub-grids with the same multiplicative grid factor. "w/ AVG" stands for polynomial iterate averaging.

remains similar for high resolutions while the gap between the fixed learning rate schedule and the decaying schedules widens for large grid factors. With a grid factor of approximately 22, the performance of the fixed learning rate decreases by 0.61, with smaller drops of 0.3, 0.35, and 0.25 observed for cosine annealing, linear decay, and quadratic decay schedules, respectively, and the trend continues for grids with lower resolutions.

### 4.3. Discussion

The experiments show that decaying schedules are more robust to coarse grids, while performance differences on fine grids remain minimal. These findings align with our theory, which suggests that all decaying schedules perform similarly to iterate averaging under small multiplicative misspecification but outperform it when misspecification is large. However, our theory also predicts robustness variations across decay rates, which are not observed in the real-data experiments. A possible explanation is the small difference in convergence rates among decaying schedules when misspecification is low, as illustrated in Figure 3 where

we compute numerically the convergence rates with different scheduling. In addition, while cosine annealing and quadratic decay have the same theoretical dependence on $\rho$ due to their matching tail behavior, tuned cosine annealing performs better in our experiments. We conjecture that the initial phase of cosine annealing, in which the learning rate decays more slowly than under quadratic decay, is beneficial for performance.

## 5. Conclusion and limitations

In this work, we explored a new perspective on the benefits of annealing schedules, namely, revealing their improved robustness to initial stepsize overestimation. This suggests that annealing schedules are more robust to coarse grid searches, a finding supported by our experimental evaluation.

A potential drawback of our theoretical analysis is its dependence on an upper bound $D$ on the distance to the minimizer $\|x_1 - x^\star\|$, rather than on the actual distance directly. While this is admittedly a limitation of our work, our result still provides meaningful insight, particularly in

settings where $\|x_1 - x^\star\|$ is large. For example, under a linear decay schedule and denoting $\eta_{\text{tu}} = D/(G\sqrt{T})$ and $\eta_{\text{tu}}^\star = \|x_1 - x^\star\|/(G\sqrt{T})$, Corollary 2 yields:

$$\mathbb{E}[f(x_{T+1}) - f(x^\star)] \leq (\eta/\eta_{\text{tu}})^{1/3} O(DG/\sqrt{T})$$
$$= (D/\|x_1 - x^\star\|)^{2/3} (\eta/\eta_{\text{tu}}^\star)^{1/3} O(\|x_1 - x^\star\|G/\sqrt{T}).$$

Thus, as long as $D/\|x_1 - x^\star\|$ is not too large relative to the misspecification with respect to $\eta_{\text{tu}}^\star$, the degradation remains sublinear, even compared to the tuned rate based on $\|x_1 - x^\star\|$. This is especially relevant under random initialization, where $\|x_1 - x^\star\|$ is typically large.

In addition, extending the perspective developed here beyond the convex setting is an important direction for future work, but doing so for general smooth non-convex optimization appears to require additional structure, as even gradient descent with any stepsize schedule need not have a stationary last iterate in this setting; see Section E.

## Acknowledgements

We are grateful to Noga Bar, Yair Carmon and Tomer Porian for helpful discussions. This project has received funding from the European Research Council (ERC) under the European Union's Horizon 2020 research and innovation program (grant agreement No. 101078075). Views and opinions expressed are however those of the author(s) only and do not necessarily reflect those of the European Union or the European Research Council. Neither the European Union nor the granting authority can be held responsible for them. This work received additional support from the Israel Science Foundation (ISF, grant number 3174/23), the Council for Higher Education in Israel under a Moonshot Project, a grant from the Tel Aviv University Center for AI and Data Science (TAD), and a fellowship from the Israeli Council for Higher Education.

## Impact Statement

This paper presents work whose goal is to advance the field of Machine Learning. There are many potential societal consequences of our work, none which we feel must be specifically highlighted here.

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

## A. Convex and smooth setting

In the following section, we extend our robustness result to the convex smooth setting, in which we replace the second-moment gradient oracle assumption with the assumptions that the gradient oracle has bounded variance and that $f$ is $\beta$-smooth. The core technique is the same as in Section 3, with some additional considerations due to the requirement in standard smooth analysis that the stepsizes satisfy $\eta_1, \ldots, \eta_T \leq \frac{c}{\beta}$ for some constant $c < 2$.

Before presenting our robustness result, we present a standard convergence guarantee in the convex smooth case with an annealed schedule in order to define the reference tuned stepsize and tuned convergence rate. For the proof see Section D.

**Lemma 4.** *Let $\mathcal{X} \subset \mathbb{R}^d$ be a convex set with diameter $D > 0$, $f : \mathcal{X} \to \mathbb{R}$ a $\beta$-smooth convex function, $x^\star \in \arg\min_{x \in \mathcal{X}} f(x)$, and $g : \mathcal{X} \to \mathbb{R}^d$ an unbiased first-order oracle of $f$ with variance bounded by $\sigma^2 \geq 0$. Let $x_1, x_2, \ldots, x_{T+1}$ be the iterates produced by $T$-steps SGD with stepsizes $\eta_t = \eta h(\frac{t-1}{T})$ using the oracle $g$, where $h$ is a differentiable $p$-Lipschitz annealed schedule and $\eta h(0) \leq \frac{1}{2\beta}$. Then it holds that*

$$\mathbb{E}[f(x_{T+1}) - f(x^\star)] \leq \frac{D^2}{2\eta T H_h(0)} + \eta \sigma^2 Q_h(0) + \frac{4p\eta\sigma^2}{T}.$$

We denote the tuned stepsize over $\eta \in (0, \frac{1}{2\beta h(0)}]$ as

$$\eta_{\text{tu}}^{\text{sm}} \triangleq \min\left\{ \frac{1}{2\beta h(0)}, \frac{D}{\sigma\sqrt{2TH_h(0)Q_h(0)}} \right\}, \tag{7}$$

and the respective convergence guarantee (up to lower-order terms) as

$$\text{Rate}_{h,T}^{\text{sm,tu}} \triangleq \frac{D^2}{2\eta_{\text{tu}}^{\text{sm}}TH_h(0)} + \eta_{\text{tu}}^{\text{sm}}\sigma^2 Q_h(0) \leq \frac{\beta D^2 h(0)}{TH_h(0)} + \frac{D\sigma}{\sqrt{T}}\sqrt{2Q_h(0)/H_h(0)}. \tag{8}$$

Next is the main result of this section, a convergence guarantee robust to a multiplicative misspecification of the stepsize.

**Theorem 4.** *Let $\mathcal{X} \subset \mathbb{R}^d$ be a convex set with diameter $D > 0$, $f : \mathcal{X} \to \mathbb{R}$ a $\beta$-smooth convex function, $x^\star \in \arg\min_{x \in \mathcal{X}} f(x)$, and $g : \mathcal{X} \to \mathbb{R}^d$ an unbiased first-order oracle of $f$ with variance bounded by $\sigma^2 \geq 0$. For any $\rho \geq 1$, let $x_1, x_2, \ldots, x_{T+1}$ be the iterates produced by $T$-steps SGD with stepsizes $\eta_t = \eta h(\frac{t-1}{T})$ using the oracle $g$, where $\eta = \rho \cdot \eta_{\text{tu}}^{\text{sm}}$ and $h$ is a differentiable $p$-Lipschitz annealed schedule. Denote $\tau_0 \triangleq \min\{\tau \in [0, 1) : \eta h(\frac{\lfloor \tau T \rfloor}{T}) \leq \frac{1}{2\beta}\}$. Then it holds that*

$$\mathbb{E}[f(x_{T+1}) - f(x^\star)] \leq \text{Rate}_{h,T}^{\text{sm,tu}} \cdot \inf_{\tau \in [\tau_0, 1)}\left( \frac{H_h(0)}{\rho H_h(\tau)} + \frac{\rho Q_h(\tau)}{Q_h(0)} \right) + O\left( \frac{p\rho\eta_{\text{tu}}^{\text{sm}}\sigma^2}{T} \right),$$

*where $H_h$, $Q_h$, $\eta_{\text{tu}}^{\text{sm}}$, and $\text{Rate}_{h,T}^{\text{sm,tu}}$ are given in Equations (1), (7) and (8). In particular, the optimal $\tau$ satisfies $H_h(\tau)H_h'(\tau) = -H_h(0)Q_h(0)/\rho^2$ (or $\tau = \tau_0$ if there is no solution).*

As in Theorem 1, Theorem 4 shows similar adaptivity to $\rho$ using the tails of $H_h$ and $Q_h$. One small yet important difference is that the infimum is limited to the range $[\tau_0, 1)$, where $\tau_0$ denotes the fraction of iterations in which the stepsize exceeds $1/(2\beta)$. This dependency is somewhat unavoidable (up to constants) as stepsizes larger or equal to $2/\beta$ do not converge. Additionally, note the above guarantee holds even if we specify a stepsize larger than $2/\beta$, which is not the case with fixed stepsize SGD.

Next are corollaries of Theorem 4 with polynomial decay and cosine annealing schedules. The proofs of the main theorem and corollaries follow.

**Corollary 5.** *In the setting of Theorem 4, let $h(u) = (1 - u)^p$ for some $p \geq 1$ and $\rho \leq \frac{T}{2p}$. Then if $\rho^2 \geq (1 - \tau_0)^{-(2p+1)}$,*

$$\mathbb{E}[f(x_{T+1}) - f(x^\star)] = \text{Rate}_{h,T}^{\text{sm}}(\eta_{\text{tu}}^{\text{sm}}) \cdot O\left( \rho^{\frac{1}{2p+1}} \right),$$

*and if $\rho^2 < (1 - \tau_0)^{-(2p+1)}$,*

$$\mathbb{E}[f(x_{T+1}) - f(x^\star)] = \text{Rate}_{h,T}^{\text{sm}}(\eta_{\text{tu}}^{\text{sm}}) \cdot O\left( \frac{1}{1 - \tau_0} \right).$$

*In addition, $\text{Rate}_{h,T}^{\text{sm,tu}} = O\left( \frac{p\beta D^2}{T} + \frac{\sqrt{p}D\sigma}{\sqrt{T}} \right).$*

**Corollary 6.** *In the setting of Theorem 4, let $h(u) = \frac{1}{2}(1 + \cos(\pi u))$ and $\rho \le \frac{2T}{\pi}$. Then if $\rho^2 \ge (1 - \tau_0)^{-5}$,*

$$\mathbb{E}[f(x_{T+1}) - f(x^\star)] = \mathsf{Rate}_{h,T}^{\mathsf{sm,tu}} \cdot O\left(\rho^{\frac{1}{5}}\right),$$

*and if $\rho^2 < (1 - \tau_0)^{-5}$,*

$$\mathbb{E}[f(x_{T+1}) - f(x^\star)] = \mathsf{Rate}_{h,T}^{\mathsf{sm,tu}} \cdot O\left(\frac{1}{1 - \tau_0}\right).$$

*In addition,* $\mathsf{Rate}_{h,T}^{\mathsf{sm,tu}} = O\left(\frac{\beta D^2}{T} + \frac{D\sigma}{\sqrt{T}}\right)$.

Observing Corollaries 5 and 6, a similar improved dependence on $\rho$ as in Corollaries 2 and 3 holds when $\tau_0$ is sufficiently small. When $\tau_0$ is large, we obtain the expected inverse dependence on the fraction of steps with small enough stepsizes, which is unavoidable as we explained above.

### A.1. Proof of Theorem 4

Let $\tau \in [\tau_0, 1)$ and let $k = \lfloor \tau T \rfloor + 1 \in [T]$. Consider the suffix $x_k, x_{k+1}, \ldots, x_{T+1}$ as an SGD sequence starting at $x_k$ and note that since $h$ is non-increasing, $\eta_k = \eta h(\frac{k-1}{T}) \le \eta h(\tau_0) \le \frac{1}{2\beta}$. Thus, by Lemma 5 with $\hat{x} = x^\star$,

$$\mathbb{E}[f(x_{T+1}) - f(x^\star)] \le \frac{D^2}{2\sum_{s=k}^T \eta_s} + \sigma^2 \sum_{t=k}^T \frac{\eta_t^2}{\sum_{s=t}^T \eta_s}.$$

As $\tau \in [\frac{k-1}{T}, \frac{k}{T})$, invoking Lemma 3 with $c_1 = D^2/2$ and $c_2 = \sigma^2$,

$$\mathbb{E}[f(x_{T+1}) - f(x^\star)] \le \frac{D^2}{2\eta T H_h(\tau)} + \eta\sigma^2 \int_\tau^1 \frac{h(u)^2}{H_h(u)} du + \frac{4\eta p \sigma^2}{T}.$$

Substituting $\eta = \rho \cdot \eta_{\mathsf{tu}}^{\mathsf{sm}}$ and using Equations (7) and (8),

$$\mathbb{E}[f(x_{T+1}) - f(x^\star)] \le \frac{1}{\rho H_h(\tau)} \cdot \frac{D^2}{2\eta_{\mathsf{tu}}^{\mathsf{sm}} T} + \left(\rho \int_\tau^1 \frac{h(u)^2}{H_h(u)} du\right) \cdot \eta_{\mathsf{tu}}^{\mathsf{sm}} \sigma^2 + \frac{4p\rho}{T} \cdot \eta_{\mathsf{tu}}^{\mathsf{sm}} \sigma^2$$

$$= \frac{H_h(0)}{\rho H_h(\tau)} \cdot \frac{D^2}{2\eta_{\mathsf{tu}}^{\mathsf{sm}} T H_h(0)} + \frac{\rho Q_h(\tau)}{Q_h(0)} \cdot \eta_{\mathsf{tu}}^{\mathsf{sm}} \sigma^2 Q_h(0) + \frac{4p\rho\eta_{\mathsf{tu}}^{\mathsf{sm}} \sigma^2}{T}$$

$$\le \mathsf{Rate}_{h,T}^{\mathsf{sm}} \cdot \left(\frac{H_h(0)}{\rho H_h(\tau)} + \frac{\rho Q_h(\tau)}{Q_h(0)}\right) + \frac{4p\rho\eta_{\mathsf{tu}}^{\mathsf{sm}} \sigma^2}{T}.$$

This inequality holds for any $\tau \in [\tau_0, 1)$, hence it holds for the infimum over all $\tau \in [\tau_0, 1)$. It is left to find the $\tau$ which minimizes the right-hand side. Let

$$g(v) = \frac{H_h(0)}{\rho H_h(\tau)} + \frac{\rho Q_h(\tau)}{Q_h(0)}.$$

This is the same function as in Equation (5), so the same solution to $H_h(\tau)H_h'(\tau) = \frac{-H_h(0)Q_h(0)}{\rho^2}$ is the minimizer of the function, and if there is no solution, the function is increasing (positive derivative) and the minimizer is at $\tau = \tau_0$. □

### A.2. Proof of Corollary 5

As in the proof of Corollary 2, $h(u)$ is *annealed* as $h(u)$ is non-increasing, differentiable, $p$-Lipschitz and satisfy $h(u) = 0 \Leftrightarrow u = 1$. Hence, we can use Theorem 4. In addition, $H_h(\tau) = \frac{1}{p+1}(1 - \tau)^{p+1}$, $H_h'(\tau) = -(1 - \tau)^p$ and $Q_h(\tau) = \frac{p+1}{p}(1 - \tau)^p$, so

$$\frac{H_h(0)}{\rho H_h(\tau)} + \frac{\rho Q_h(\tau)}{Q_h(0)} = \frac{1}{\rho(1 - \tau)^{p+1}} + \rho(1 - \tau)^p.$$

If $\rho^2 \geq (1 - \tau_0)^{-(2p+1)}$ we can pick $\bar{\tau} = 1 - \rho^{\frac{-2}{2p+1}}$, as $\bar{\tau} \in [\tau_0, 1)$. In this case,

$$\frac{H_h(0)}{\rho H_h(\bar{\tau})} + \frac{\rho Q_h(\bar{\tau})}{Q_h(0)} = \frac{1}{\rho \cdot \rho^{\frac{-2(p+1)}{2p+1}}} + \rho \cdot \rho^{\frac{-2p}{2p+1}} = 2\rho^{\frac{1}{2p+1}}.$$

If $\rho^2 < (1 - \tau_0)^{-(2p+1)}$ and $\tau_0 > 0$, picking $\bar{\tau} = \tau_0$ and using the $p$-Lipschitz property of $h$,

$$\frac{1}{2\beta} \leq \eta h\left(\tau_0 - \frac{1}{T}\right) = \rho \eta_{\mathsf{tu}}^{\mathsf{sm}} h\left(\tau_0 - \frac{1}{T}\right) \leq \rho \eta_{\mathsf{tu}}^{\mathsf{sm}} h(\tau_0) + \frac{p\rho\eta_{\mathsf{tu}}^{\mathsf{sm}}}{T} \leq \frac{\rho h(\tau_0)}{2\beta h(0)} + \frac{p\rho}{2\beta h(0)T} \qquad (\eta_{\mathsf{tu}}^{\mathsf{sm}} h(0) \leq \frac{1}{2\beta})$$

$$\implies \rho \geq \left(1 - \frac{p\rho}{T}\right)(1 - \tau_0)^{-p} \qquad\qquad (h(0) = 1)$$

and

$$\frac{H_h(0)}{\rho H_h(\tau_0)} + \frac{\rho Q_h(\tau_0)}{Q_h(0)} = \frac{1}{\rho(1-\tau_0)^{p+1}} + \rho(1-\tau_0)^p \leq \frac{1}{\left(1 - \frac{p\rho}{T}\right)(1-\tau_0)} + \sqrt{\frac{1}{1-\tau_0}} = O\left(\frac{1}{1-\tau_0}\right),$$

where the last two transitions use $\rho^2 < (1 - \tau_0)^{-(2p+1)}$ and the assumption $\rho \leq \frac{T}{2p}$. Since $\rho \geq 1$ there is no case where $\rho^2 < (1 - \tau_0)^{-(2p+1)}$ and $\tau_0 = 0$. Bounding the infimum of Equation ($\eta_{\mathsf{tu}}^{\mathsf{sm}} h(0) \leq \frac{1}{2\beta}$) in the two cases with our choices of $\bar{\tau}$, if $\rho^2 \geq (1 - \tau_0)^{-(2p+1)}$,

$$\mathbb{E}[f(x_{T+1}) - f(x^\star)] = \mathsf{Rate}_{h,T}^{\mathsf{sm,tu}} \cdot O\left(\rho^{\frac{1}{2p+1}}\right) + O\left(\frac{p\rho\eta_{\mathsf{tu}}^{\mathsf{sm}}\sigma^2}{T}\right),$$

and if $\rho^2 < (1 - \tau_0)^{-(2p+1)}$,

$$\mathbb{E}[f(x_{T+1}) - f(x^\star)] = \mathsf{Rate}_{h,T}^{\mathsf{sm,tu}} \cdot O\left(\frac{1}{1-\tau_0}\right) + O\left(\frac{p\rho\eta_{\mathsf{tu}}^{\mathsf{sm}}\sigma^2}{T}\right).$$

Noting that by the assumption $\rho \leq \frac{T}{2p}$,

$$\frac{p\rho\eta_{\mathsf{tu}}^{\mathsf{sm}}\sigma^2}{T} \leq \frac{\eta_{\mathsf{tu}}^{\mathsf{sm}}\sigma^2}{2} \leq \frac{\mathsf{Rate}_{h,T}^{\mathsf{sm,tu}}}{2Q_h(0)} = O(\mathsf{Rate}_{h,T}^{\mathsf{sm,tu}}),$$

we obtain our final convergence guarantees. The bound of $\mathsf{Rate}_{h,T}^{\mathsf{sm,tu}}$ follows from plugging $H_h(0) = \frac{1}{p+1}$ and $Q_h(0) = \frac{p+1}{p}$ to Equation (8). $\qquad\square$

### A.3. Proof of Corollary 6

As in the proof of Corollary 3, $h$ is *annealed* as $h(u)$ is non-increasing, differentiable, $\frac{\pi}{2}$-Lipschitz and satisfy $h(u) = 0 \Leftrightarrow u = 1$. Hence, we can use Theorem 4. We already established at Equation (19) of the proof of Corollary 3 that

$$\frac{H_h(0)}{\rho H_h(\tau)} + \frac{\rho Q_h(\tau)}{Q_h(0)} \leq \frac{3}{2\rho(1-\tau)^3} + \frac{125\rho(1-\tau)^2}{8}.$$

If $\rho^2 \geq (1 - \tau_0)^{-5}$ we can pick $\bar{\tau} = 1 - \rho^{\frac{-2}{5}}$, as $\bar{\tau} \in [\tau_0, 1)$. In this case,

$$\frac{H_h(0)}{\rho H_h(\bar{\tau})} + \frac{\rho Q_h(\bar{\tau})}{Q_h(0)} \leq \frac{3}{2\rho \cdot \rho^{\frac{-6}{5}}} + \frac{125\rho \cdot \rho^{\frac{-4}{5}}}{8} = \frac{137\rho^{\frac{1}{5}}}{8} \leq 18\rho^{\frac{1}{5}}.$$

If $\rho^2 < (1 - \tau_0)^{-5}$ and $\tau_0 > 0$, picking $\bar{\tau} = \tau_0$, using the definition of $\tau_0$ and the Lipschitz property of $h$,

$$\frac{1}{2\beta} \leq \eta h\left(\tau_0 - \frac{1}{T}\right) = \rho \eta_{\mathsf{tu}}^{\mathsf{sm}} h\left(\tau_0 - \frac{1}{T}\right) \leq \rho \eta_{\mathsf{tu}}^{\mathsf{sm}} h(\tau_0) + \frac{\pi\rho\eta_{\mathsf{tu}}^{\mathsf{sm}}}{2T} \leq \frac{\rho h(\tau_0)}{2\beta h(0)} + \frac{\pi\rho}{4\beta h(0)T}, \qquad (\eta_{\mathsf{tu}}^{\mathsf{sm}} h(0) \leq \frac{1}{2\beta})$$

implying (with $h(0) = 1$) that

$$\rho \geq \left(1 - \frac{\pi\rho}{2T}\right)h(\tau_0) \geq \left(1 - \frac{\pi\rho}{2T}\right)(1 - \tau_0)^2.$$

In addition to the assumption $\rho^2 < (1 - \tau_0)^{-5}$,

$$\frac{H_h(0)}{\rho H_h(\tau_0)} + \frac{\rho Q_h(\tau_0)}{Q_h(0)} \leq \frac{3}{2\rho(1 - \tau_0)^3} + \frac{125\rho(1 - \tau_0)^2}{8}$$

$$\leq \frac{3}{2\left(1 - \frac{\pi\rho}{2T}\right)(1 - \tau_0)} + \frac{125}{8}\sqrt{\frac{1}{1 - \tau_0}} = O\left(\frac{1}{1 - \tau_0}\right),$$

where the last transition uses the assumption $\rho \leq \frac{2T}{\pi}$. Since $\rho \geq 1$ there is no case where $\rho^2 < (1 - \tau_0)^{-(2p+1)}$ and $\tau_0 = 0$. Bounding the infimum of Equation ($\eta_{\text{tu}}^{\text{sm}}h(0) \leq \frac{1}{2\beta}$) in the two cases with our choices of $\bar\tau$, if $\rho^2 \geq (1 - \tau_0)^{-5}$,

$$\mathbb{E}[f(x_{T+1}) - f(x^\star)] = \text{Rate}_{h,T}^{\text{sm,tu}} \cdot O\left(\rho^{\frac{1}{5}}\right) + O\left(\frac{\rho\eta_{\text{tu}}^{\text{sm}}\sigma^2}{T}\right),$$

and if $\rho^2 < (1 - \tau_0)^{-5}$,

$$\mathbb{E}[f(x_{T+1}) - f(x^\star)] = \text{Rate}_{h,T}^{\text{sm,tu}} \cdot O\left(\frac{1}{1 - \tau_0}\right) + O\left(\frac{\rho\eta_{\text{tu}}^{\text{sm}}\sigma^2}{T}\right).$$

We obtain our final convergence guarantees by noting that $\rho \leq \frac{2T}{\pi}$, which, together with the fact that $Q_h(0) = \Theta(1)$ implies

$$\frac{\rho\eta_{\text{tu}}^{\text{sm}}\sigma^2}{T} \leq \frac{2\eta_{\text{tu}}^{\text{sm}}\sigma^2}{\pi} \leq \frac{2\text{Rate}_{h,T}^{\text{sm,tu}}}{\pi Q_h(0)} = O(\text{Rate}_{h,T}^{\text{sm,tu}}),$$

and plugging back to the above bounds. The bound of $\text{Rate}_{h,T}^{\text{sm,tu}}$ is immediate from Equation (8) as $H_h(0) = \frac{1}{2}$ and $Q_h(0) = \Theta(1)$ (as we established in Equation (18)).

$\square$

## B. Last iterate guarantees for stochastic gradient descent

A convergence analysis of Stochastic Gradient Descent (SGD) for convex Lipschitz and convex smooth functions follows. The technique, introduced by Zamani & Glineur (2025) and later refined by Liu & Zhou (2024), is based on comparing the iterates of SGD $(x_1, x_2, \ldots)$ with iterates of the form

$$z_t \triangleq \frac{v_{t-1}}{v_t}z_{t-1} + \left(1 - \frac{v_{t-1}}{v_t}\right)x_t = \frac{v_0}{v_t}\hat{x} + \sum_{s=1}^{t}\frac{v_s - v_{s-1}}{v_t}x_s \tag{9}$$

for some non-increasing sequence $v_0, v_1, v_2, \ldots$, starting at some $z_0 = \hat{x} \in \mathcal{X}$. Note that by Jensen's inequality, for any $t \geq 2$,

$$f(z_t) \leq \frac{v_0}{v_t}f(\hat{x}) + \sum_{s=1}^{t}\frac{v_s - v_{s-1}}{v_t}f(x_s). \tag{10}$$

In particular, for any $t \in [T]$, we will use

$$v_t \triangleq \frac{\eta_T}{\sum_{s=t}^{T}\eta_s} \tag{11}$$

and $v_0 = v_1$, similarly to Liu & Zhou (2024). Next, we restate the convergence results. Their proofs follow.

**Lemma 2.** *Let $X \subseteq \mathbb{R}^d$ be a convex set, $f : X \to \mathbb{R}$ a convex function, and $g : X \to \mathbb{R}^d$ an unbiased first-order oracle of $f$ with second-moment bounded by $G^2 > 0$. Let $x_1, \ldots, x_{T+1}$ be the iterates produced by $T$-steps SGD with stepsizes $\eta_1, \ldots, \eta_T$ using the oracle $g$. Then for any $\hat{x} \in X$,*

$$\mathbb{E}[f(x_{T+1}) - f(\hat{x})] \leq \frac{\|x_1 - \hat{x}\|^2}{2 \sum_{s=1}^T \eta_s} + 2G^2 \sum_{t=1}^T \frac{\eta_t^2}{\sum_{s=t}^T \eta_s}.$$

**Lemma 5.** *Let $X \subseteq \mathbb{R}^d$ be a convex set, $f : X \to \mathbb{R}$ a convex function, and $g : X \to \mathbb{R}^d$ an unbiased first-order oracle of $f$ with variance bounded by $\sigma^2 \geq 0$. Let $x_1, x_2, \ldots, x_{T+1}$ be the iterates produced by $T$-steps SGD with stepsizes $\eta_1, \ldots, \eta_T$ (satisfying $\eta_t \leq \frac{1}{2\beta}$ for all $t \in [T]$) and using the oracle $g$. Then for any $\hat{x} \in X$,*

$$\mathbb{E}[f(x_{T+1}) - f(\hat{x})] \leq \frac{\|x_1 - \hat{x}\|^2}{2 \sum_{s=1}^T \eta_s} + \sigma^2 \sum_{t=1}^T \frac{\eta_t^2}{\sum_{s=t}^T \eta_s}.$$

### B.1. Proof of Lemmas 2 and 5

To prove the last-iterate guarantees we need the following lemmas. Their proofs follow. The first translates from an average regret-like guarantee to a last-iterate guarantee.

**Lemma 6.** *Let $X \subseteq \mathbb{R}^d$ be a convex set, $x_1, \hat{x} \in X$, $f : X \to \mathbb{R}$ a convex function and $T \in \mathbb{N}$. Then for any sequences $g_1, \ldots, g_T \in \mathbb{R}^d$ and $\eta_1, \ldots, \eta_T > 0$, the iterates defined by $x_{t+1} = x_t - \eta_t g_t$ satisfy*

$$\eta_T v_T (f(x_{T+1}) - f(\hat{x})) \leq \sum_{t=1}^T \eta_t v_t (f(x_{t+1}) - f(z_t)),$$

*where $z_0, \ldots, z_T$ and $v_0, \ldots, v_T$ are defined by Equations (9) and (11).*

**Lemma 7.** *Let $X \subseteq \mathbb{R}^d$ be a convex set, $x_1, \hat{x} \in X$, $f : X \to \mathbb{R}$ a convex function and $T \in \mathbb{N}$. Then for any $t \in [T]$, the iterates of SGD satisfy*

$$\mathbb{E}[f(x_{t+1}) - f(z_t)] \leq \mathbb{E}\left[ \frac{\|x_t - z_t\|^2 - \|x_{t+1} - z_t\|^2 - \|x_{t+1} - x_t\|^2}{2\eta_t} + f(x_{t+1}) - f(x_t) + g_t \cdot (x_t - x_{t+1}) \right],$$

*where $z_0, \ldots, z_T$ are defined by Equation (9).*

We proceed to the proof.

*Proof of Lemmas 2 and 5.* By Lemma 7,

$$\mathbb{E}[f(x_{t+1}) - f(z_t)] \leq \mathbb{E}\left[ \frac{\|x_t - z_t\|^2 - \|x_{t+1} - z_t\|^2}{2\eta_t} + \Delta_t \right],$$

where $\Delta_t \triangleq f(x_{t+1}) - f(x_t) + g_t \cdot (x_t - x_{t+1}) - \frac{\|x_{t+1} - x_t\|^2}{2\eta_t}$. By the definition of $z_t$ and the fact that $v_t \geq v_{t-1}$,

$$\|x_t - z_t\|^2 = \frac{v_{t-1}^2}{v_t^2} \|x_t - z_{t-1}\|^2 \leq \frac{v_{t-1}}{v_t} \|x_t - z_{t-1}\|^2.$$

Combining with our previous inequality multiplied by $\eta_t v_t$,

$$\mathbb{E}[\eta_t v_t (f(x_{t+1}) - f(z_t))] \leq \mathbb{E}\left[ \frac{v_{t-1}\|x_t - z_{t-1}\|^2 - v_t\|x_{t+1} - z_t\|^2}{2} + \eta_t v_t \Delta_t \right].$$

Summing for $t = 1, \ldots, T$, and removing $-v_T\|x_{T+1} - z_T\|^2 \leq 0$,

$$\mathbb{E}\left[ \sum_{t=1}^T \eta_t v_t (f(x_{t+1}) - f(z_t)) \right] \leq \mathbb{E}\left[ \frac{v_0\|x_1 - z_0\|^2}{2} + \sum_{t=1}^T \eta_t v_t \Delta_t \right].$$

Combining with Lemma 6, and noting that $z_0 = \hat{x}$,

$$\eta_T v_T (f(x_{T+1}) - f(\hat{x})) \leq \mathbb{E}\left[\frac{v_0\|x_1 - \hat{x}\|^2}{2} + \sum_{t=1}^{T} \eta_t v_t \Delta_t\right]. \tag{12}$$

Next, we assume a second-moment bound (as in Lemma 2). From convexity,

$$\mathbb{E}[f(x_{t+1}) - f(x_t)] \leq \mathbb{E}[\nabla f(x_{t+1}) \cdot (x_{t+1} - x_t)] \leq \mathbb{E}\left[\eta_t \|\nabla f(x_t)\|^2 + \frac{\|x_{t+1} - x_t\|^2}{4\eta_t}\right],$$

where we used the inequality $2u \cdot v \leq \|u\|^2 + \|v\|^2$. Similarly, $g_t \cdot (x_t - x_{t+1}) \leq \eta_t \|g_t\|^2 + \frac{\|x_{t+1} - x_t\|}{4\eta_t}$. Hence, using the second-moment bound, $\mathbb{E}\Delta_t \leq 2\eta_t G^2$. Plugging the bound of $\mathbb{E}[\Delta_t]$ to Equation (12) concludes the proof of Lemma 2. Next we assume that $f$ is $\beta$-smooth, a variance bound, and that $\eta_t \leq \frac{1}{2\beta}$ for all $t \in [T]$ (as in Lemma 5). By smoothness,

$$f(x_{t+1}) - f(x_t) \leq \nabla f(x_t) \cdot (x_{t+1} - x_t) + \frac{\beta}{2}\|x_{t+1} - x_t\|^2$$

$$\leq \nabla f(x_t) \cdot (x_{t+1} - x_t) + \frac{1}{4\eta_t}\|x_{t+1} - x_t\|^2. \tag{$\eta_t \leq \frac{1}{2\beta}$}$$

By the inequality $2u \cdot v \leq \|u\|^2 + \|v\|^2$,

$$\nabla f(x_t) \cdot (x_{t+1} - x_t) = (\nabla f(x_t) - g_t) \cdot (x_{t+1} - x_t) + g_t \cdot (x_{t+1} - x_t)$$

$$\leq \eta_t \|\nabla f(x_t) - g_t\|^2 + \frac{\|x_{t+1} - x_t\|^2}{4\eta_t} + g_t \cdot (x_{t+1} - x_t).$$

Hence, using the variance bound,

$$\mathbb{E}\Delta_t \leq \mathbb{E}\left[\eta_t \|\nabla f(x_t) - g_t\|^2 + \frac{\|x_{t+1} - x_t\|^2}{4\eta_t} + \frac{\|x_{t+1} - x_t\|^2}{4\eta_t} - \frac{\|x_{t+1} - x_t\|^2}{2\eta_t}\right] \leq \eta_t \sigma^2.$$

Plugging the bound of $\mathbb{E}[\Delta_t]$ to Equation (12) concludes the proof of Lemma 5. $\qquad\square$

### B.2. Proof of Lemma 6

*Proof.* By Equation (10),

$$\sum_{t=1}^{T} \eta_t v_t (f(x_{t+1}) - f(z_t)) \geq \sum_{t=1}^{T} \eta_t v_t \left(f(x_{t+1}) - \left(\frac{v_0}{v_t}f(\hat{x}) + \sum_{s=1}^{t} \frac{v_s - v_{s-1}}{v_t}f(x_s)\right)\right)$$

$$= \sum_{t=1}^{T} \eta_t \left(v_t f(x_{t+1}) - \left(v_0 f(\hat{x}) + \sum_{s=1}^{t}(v_s - v_{s-1})f(x_s)\right)\right)$$

$$= \sum_{t=1}^{T} \eta_t \left(v_t f(x_{t+1}) - v_t f(\hat{x}) - \sum_{s=1}^{t}(v_s - v_{s-1})(f(x_s) - f(\hat{x}))\right)$$

$$= \eta_T v_T (f(x_{T+1}) - f(\hat{x}))$$

$$+ \sum_{t=1}^{T}\left(\eta_{t-1}v_{t-1} - (v_t - v_{t-1})\sum_{s=t}^{T}\eta_s\right)(f(x_t) - f(\hat{x})).$$

Note that $v_1 = v_0$ and for $2 \leq t \leq T$,

$$\eta_{t-1}v_{t-1} - (v_t - v_{t-1})\sum_{s=t}^{T}\eta_s = \eta_T\left(\frac{\eta_{t-1}}{\sum_{s=t-1}^{T}\eta_s} - \frac{\eta_{t-1}\sum_{s=t}^{T}\eta_s}{\sum_{s=t-1}^{T}\eta_s \cdot \sum_{s=t}^{T}\eta_s}\right) = 0.$$

Thus,

$$\eta_T v_T (f(x_{T+1}) - f(\hat{x})) \leq \sum_{t=1}^{T} \eta_t v_t (f(x_{t+1}) - f(z_t)). \qquad\square$$

### B.3. Proof of Lemma 7

*Proof.* Using the convexity of $f$,

$$
\begin{aligned}
\mathbb{E}[f(x_{t+1}) - f(z_t)] &= \mathbb{E}[f(x_t) - f(z_t) + f(x_{t+1}) - f(x_t)] \\
&\leq \mathbb{E}[\nabla f(x_t) \cdot (x_t - z_t) + f(x_{t+1}) - f(x_t)].
\end{aligned}
\tag{13}
$$

Focusing on the first term, as $z_t$ does not depend on $g_t$,

$$
\mathbb{E}[\nabla f(x_t) \cdot (x_t - z_t)] = \mathbb{E}[g_t \cdot (x_t - z_t)] = \mathbb{E}[g_t \cdot (x_{t+1} - z_t) + g_t \cdot (x_t - x_{t+1})].
$$

Note that the update step is

$$
x_{t+1} = \arg\min_{x \in \mathcal{X}} \left\{ f(x_t) + g_t \cdot (x - x_t) + \frac{1}{2\eta_t} \|x - x_t\|^2 \right\}.
$$

From the first-order optimality condition,

$$
\frac{1}{\eta_t}(x_{t+1} - x_t + \eta_t g_t) \cdot (z_t - x_{t+1}) \geq 0.
$$

Rearranging,

$$
g_t \cdot (x_{t+1} - z_t) \leq \frac{\|x_t - z_t\|^2 - \|x_{t+1} - z_t\|^2 - \|x_{t+1} - x_t\|^2}{2\eta_t}.
$$

Thus,

$$
\mathbb{E}[\nabla f(x_t) \cdot (x_t - z_t)] \leq \mathbb{E}\left[ \frac{\|x_t - z_t\|^2 - \|x_{t+1} - z_t\|^2 - \|x_{t+1} - x_t\|^2}{2\eta_t} + g_t \cdot (x_t - x_{t+1}) \right].
$$

Returning to Equation (13), we conclude that

$$
\mathbb{E}[f(x_{t+1}) - f(z_t)] \leq \mathbb{E}\left[ \frac{\|x_t - z_t\|^2 - \|x_{t+1} - z_t\|^2 - \|x_{t+1} - x_t\|^2}{2\eta_t} + f(x_{t+1}) - f(x_t) + g_t \cdot (x_t - x_{t+1}) \right].
$$

$\square$

## C. Sensitivity of non-annealing schedules to misspecification of the stepsize

### C.1. Sensitivity with a fixed stepsize

Given a $G$-Lipschitz function $f : \mathcal{X} \to \mathbb{R}$, where $\mathcal{X} \subset \mathbb{R}^d$ is a convex set with diameter $D$, the standard average-iterate convergence guarantee of $T$-steps Gradient Descent (GD) with a fixed stepsize $\eta > 0$ is

$$
f\left( \frac{1}{T} \sum_{t=1}^{T} x_t \right) - \min_{x \in \mathcal{X}} f(x) \leq \mathsf{Rate}_{\mathrm{con},T}(\eta) \triangleq \frac{D^2}{2\eta T} + \frac{\eta G^2}{2}.
$$

The optimal $\eta_{\mathrm{tu}} = \frac{D}{G\sqrt{T}}$ satisfy $\mathsf{Rate}_{\mathrm{con},T}(\eta_{\mathrm{tu}}) = \frac{DG}{\sqrt{T}}$. Given a multiplicative overestimation of the optimal stepsize, $\eta = \rho \eta_{\mathrm{tu}}$ for $\rho \geq 1$, the convergence guarantee is

$$
\mathsf{Rate}_{\mathrm{con},T}(\rho \eta_{\mathrm{tu}}) = \mathsf{Rate}_{\mathrm{con},T}(\eta_{\mathrm{tu}}) \left( \frac{1}{2\rho} + \frac{\rho}{2} \right) = \Omega(\rho \mathsf{Rate}_{\mathrm{con},T}(\eta_{\mathrm{tu}})).
$$

A natural follow-up question is whether this linear dependence on $\rho$ is simply an artifact of the analysis or a true degradation in the convergence rate of GD. Next, we show that for any weights $w_1, \ldots, w_T$, the worst-case convergence rate of the (weighted) average iterate is $\Omega(\rho \mathsf{Rate}_{\mathrm{con},T}(\eta_{\mathrm{tu}}))$.

Let $T \in \mathbb{N}$, $D > 0$, $G > 0$, $0 < \rho < \frac{1}{2}\sqrt{T}$ and $w_1, \ldots, w_T \geq 0$ such that $\sum_{t=1}^{T} w_t > 0$. First we will assume that $w_1 + w_3 + \ldots + w_{2\lfloor (T-1)/2 \rfloor + 1} \geq w_2 + w_4 + \ldots + w_{2\lfloor T/2 \rfloor}$. Let $\eta = \frac{\rho D}{G\sqrt{T}}$ for some $\rho \geq 1$, $f(x) = G|x|$ defined over the domain $\mathcal{X} = [-\frac{D}{2}, \frac{D}{2}]$, and let $x_1 = \frac{3}{4}G\eta \in (0, \frac{3}{8}D)$ (point inside the domain). After a single gradient step, $x_2 = \Pi_{\mathcal{X}}(x_1 - \eta G) = -\frac{1}{4}G\eta \in (-\frac{1}{8}D, 0)$. After another update step, $x_3 = x_2 + \eta G = \frac{3}{4}G\eta = x_1$. Hence, the iterates will move back and forth between $\frac{3}{4}G\eta$ and $-\frac{1}{4}G\eta$, and the average iterate $\bar{x}$ will satisfy

$$\bar{x} = \frac{1}{\sum_{t=1}^{T} w_t} \sum_{t=1}^{T} w_t x_t = \frac{G\eta\left(3\sum_{t=1,3,\ldots} w_t - \sum_{t=2,4,\ldots} w_t\right)}{4\sum_{t=1}^{T} w_t} \geq \frac{G\eta\left(2\sum_{t=1,3,\ldots} w_t\right)}{8\sum_{t=1,3,\ldots} w_t} = \frac{\eta G}{4},$$

where we used our assumption that $w_1 + w_3 + \ldots + w_{2\lfloor (T-1)/2 \rfloor + 1} \geq w_2 + w_4 + \ldots + w_{2\lfloor T/2 \rfloor}$. Hence,

$$f(\bar{x}) \geq \frac{\eta G^2}{4} = \frac{\rho D G}{4\sqrt{T}} = \Omega(\rho \mathsf{Rate}_{\mathrm{con},T}(\eta_{\mathrm{tu}})).$$

If, on the other hand, it holds that $w_1 + w_3 + \ldots + w_{2\lfloor (T-1)/2 \rfloor + 1} < w_2 + w_4 + \ldots + w_{2\lfloor T/2 \rfloor}$, we can initialize $x_1 = -\frac{G\eta}{4}$ and mirroring the same argument will conclude the proof.

Hence, the worst-case convergence rate of fixed stepsize GD degrades linearly in a multiplicative misspecification of the stepsize. As GD is a special case of SGD, the lower bound also holds for SGD with a second-moment bound $G^2$.

### C.2. Sensitivity of the inverse-square-root schedule

Next is an example showing that even for the natural "any-time" schedule $\eta_t = \eta/\sqrt{t}$, infinitely many iterates suffer (in the worst case) a linear dependence on the misspecification parameter $\rho$, with respect to the tuned stepsizes $\eta_t = D/(G\sqrt{t})$ when minimizing a $G$-Lipschitz convex function $f$ over a domain with diameter $D$.

Note that by Lemma 2, the tuned schedule produces a nearly-optimal last-iterate convergence guarantee,

$$\mathbb{E}[f(x_{T+1}) - f(\hat{x})] \leq \frac{DG}{2\sum_{s=1}^{T} 1/\sqrt{s}} + 2DG \sum_{t=1}^{T} \frac{1/t}{\sum_{s=t}^{T} 1/\sqrt{s}}$$

$$\leq \frac{DG}{4(\sqrt{T+1} - 1)} + DG \sum_{t=1}^{T} \frac{1/t}{\sqrt{T+1} - \sqrt{t}} \qquad \text{(Bounding sum by integration)}$$

$$\leq \frac{DG}{4(\sqrt{T+1} - 1)} + DG \sum_{t=1}^{T} \frac{2\sqrt{T+1}}{t(T+1-t)}$$

$$= \frac{DG}{4(\sqrt{T+1} - 1)} + \frac{2DG}{\sqrt{T+1}} \sum_{t=1}^{T} \left(\frac{1}{t} + \frac{1}{T+1-t}\right) = O\left(\frac{DG\ln(T+1)}{\sqrt{T}}\right). \qquad (\sum_{n=1}^{N} \frac{1}{n} \leq 1 + \ln(N))$$

Let $f(x) = G|x|$ (which is a convex $G$-Lipschitz function over a domain of diameter $D$). Let $x_1 = \frac{1}{2}D$ and $\eta_t = \rho D/G\sqrt{t}$ for $\rho \in (1, \sqrt{T/16})$. The gradient descent update step is the Euclidean projection of $x_t - \eta_t g_t$, where $g_t$ is a sub-gradient at $x_t$, and for simplicity assume we use $+G$ as the sub-gradient at 0.

Let $T_0 = \lceil 4\rho^2 \rceil \leq T/4$, such that for all $t \geq T_0$, $\eta_t \leq 0.5D/G$, which means that the projection from this point will be the identity as

$$|x_t - \eta_t g_t| = ||x_t| - \eta_t G| \leq \max\{|x_t|, \eta_t G\} \leq \frac{1}{2}D.$$

Hence, by the triangle inequality, for any $t \geq T_0$, $|x_t| + |x_{t+1}| \geq G\eta_t$. Summing for $t = T_0, \ldots, T-1$,

$$|x_{T_0}| + |x_T| + 2\sum_{t=T_0+1}^{T-1} |x_t| \geq G\sum_{t=T_0}^{T-1} \eta_t \geq 2\rho D(\sqrt{T} - \sqrt{T_0}).$$

Hence, after rearranging and adding non-negative terms,

$$\frac{1}{T} \sum_{t=1}^{T} f(x_t) \geq DG \frac{\rho(\sqrt{T} - \sqrt{T_0})}{T} = \Omega\left(\frac{\rho DG}{\sqrt{T}}\right).$$

This implies that infinitely many iterates must satisfy $f(x_t) \geq \Omega(\frac{\rho DG}{\sqrt{t}})$, and that the sub-optimality of a uniformly random iterate $\bar{x}$ of $T$-steps SGD satisfies $\mathbb{E}[f(\bar{x})] \geq \Omega(\frac{\rho DG}{\sqrt{T}})$.

## D. Convergence analysis with stepsize schedules

In this section, we provide convergence guarantees for SGD with an annealed schedule in the convex Lipschitz and convex smooth settings. The guarantees are established by combining a last-iterate guarantee with Lemma 3, which translates the sums of stepsizes to integrals that depend on the schedule. The proofs follow.

**Lemma 1.** *Let $X \subset \mathbb{R}^d$ be a convex set with diameter $D > 0$, $f : X \to \mathbb{R}$ a convex function, $x^\star \in \arg\min_{x \in X} f(x)$, and $g : X \to \mathbb{R}^d$ an unbiased first-order oracle of $f$ with second-moment bounded by $G^2 > 0$. Let $x_1, x_2, \ldots, x_{T+1}$ be the iterates produced by $T$-steps SGD with stepsizes $\eta_t = \eta h(\frac{t-1}{T})$ using the oracle $g$, where $h$ is a differentiable $p$-Lipschitz annealed schedule. Then it holds that*

$$\mathbb{E}[f(x_{T+1}) - f(x^\star)] \leq \frac{D^2}{2\eta T H_h(0)} + 2\eta G^2 Q_h(0) + \frac{8p\eta G^2}{T}.$$

Note that when we tune $\eta$ according to Equation (2), we obtain a convergence rate of

$$\frac{2DG}{\sqrt{T}}\sqrt{Q_h(0)/H_h(0)} + O\left(\frac{pDG/\sqrt{H_h(0)Q_h(0)}}{T^{3/2}}\right).$$

**Lemma 4.** *Let $X \subset \mathbb{R}^d$ be a convex set with diameter $D > 0$, $f : X \to \mathbb{R}$ a $\beta$-smooth convex function, $x^\star \in \arg\min_{x \in X} f(x)$, and $g : X \to \mathbb{R}^d$ an unbiased first-order oracle of $f$ with variance bounded by $\sigma^2 \geq 0$. Let $x_1, x_2, \ldots, x_{T+1}$ be the iterates produced by $T$-steps SGD with stepsizes $\eta_t = \eta h(\frac{t-1}{T})$ using the oracle $g$, where $h$ is a differentiable $p$-Lipschitz annealed schedule and $\eta h(0) \leq \frac{1}{2\beta}$. Then it holds that*

$$\mathbb{E}[f(x_{T+1}) - f(x^\star)] \leq \frac{D^2}{2\eta T H_h(0)} + \eta\sigma^2 Q_h(0) + \frac{4p\eta\sigma^2}{T}.$$

Similarly, when we tune $\eta$ according to Equation (7), we obtain a convergence rate of

$$\frac{\beta D^2 h(0)}{T H_h(0)} + \frac{D\sigma}{\sqrt{T}}\sqrt{2Q_h(0)/H_h(0)} + O\left(\frac{pD\sigma/\sqrt{H_h(0)Q_h(0)}}{T^{3/2}}\right).$$

Note that using the fact that $h$ is non-increasing and the Lipschitz condition,

$$h(0) \geq H_h(0) = \int_0^1 h(u)du \geq \int_0^{\min\{1,h(0)/2p\}} \frac{1}{2}h(0)du = \frac{1}{2}h(0)\min\{1, h(0)/2p\}.$$

Additionally,

$$Q_h(0) = \int_0^1 \frac{h(u)^2}{H_h(u)}du \geq \int_0^1 h(u)du = H_h(0) \geq \frac{1}{2}h(0)\min\{1, h(0)/2p\}$$

and using Equation (6),

$$Q_h(0) = \int_0^1 \frac{H_h'(u)^2}{H_h(u)}du \leq 2p.$$

Hence, assuming $h(0) = \Theta(1)$ and $p = \Theta(1)$, $H_h(0)$ and $Q_h(0)$ are $\Theta(1)$, and the rates above match those of optimally tuned fixed stepsize SGD up to constant factors.

### D.1. Proofs of Lemmas 1 and 4

*Proof of Lemma 1.* By Lemma 2 with $\hat{x} = x^{\star}$,

$$\mathbb{E}[f(x_{T+1}) - f(x^{\star})] \leq \frac{D^2}{2 \sum_{s=1}^{T} \eta_s} + 2G^2 \sum_{t=1}^{T} \frac{\eta_t^2}{\sum_{s=t}^{T} \eta_s}.$$

Using Lemma 3 with $c_1 = D^2/2$, $c_2 = 2G^2$, $k = 1$ and $\tau = 0$,

$$\mathbb{E}[f(x_{T+1}) - f(x^{\star})] \leq \frac{D^2}{2\eta T H_h(0)} + 2\eta G^2 \int_0^{1-\frac{1}{T}} \frac{h(u)^2}{H_h(u)} du + \frac{8p\eta G^2}{T}$$

$$\leq \frac{D^2}{2\eta T H_h(0)} + 2\eta G^2 Q_h(0) + \frac{8p\eta G^2}{T},$$

where that last inequality follows by the fact that $h(u)$ and $H_h(u)$ are non-negative and the definition of $Q_h(u)$. $\qquad\square$

*Proof of Lemma 4.* As $\eta_1 = \eta h(0) \leq \frac{1}{2\beta}$ and $h$ is non-increasing, $\eta_t \leq \frac{1}{2\beta}$ and we can use Lemma 5 with $\hat{x} = x^{\star}$, obtaining

$$\mathbb{E}[f(x_{T+1}) - f(x^{\star})] \leq \frac{D^2}{2 \sum_{s=k}^{T} \eta_s} + \sigma^2 \sum_{t=k}^{T} \frac{\eta_t^2}{\sum_{s=t}^{T} \eta_s}.$$

Invoking Lemma 3 with $c_1 = D^2/2$, $c_2 = \sigma^2$, $k = 1$ and $\tau = 0$,

$$\mathbb{E}[f(x_{T+1}) - f(x^{\star})] \leq \frac{D^2}{2\eta T H_h(0)} + \eta \sigma^2 \int_0^{1-\frac{1}{T}} \frac{h(u)^2}{H_h(u)} du + \frac{4\eta p \sigma^2}{T}$$

$$\leq \frac{D^2}{2\eta T H_h(0)} + \eta \sigma^2 Q_h(0) + \frac{4p\eta \sigma^2}{T},$$

where that last inequality follows by the fact that $h(u)$ and $H_h(u)$ are non-negative and the definition of $Q_h(u)$. $\qquad\square$

## E. A last-iterate impossibility for gradient descent in smooth non-convex optimization

We give a simple example showing that, in general non-convex smooth optimization, one cannot hope for a last-iterate stationarity guarantee for gradient descent without additional assumptions. Fix any horizon $T \in \mathbb{N}$ and any prescribed stepsizes $\eta_1, \ldots, \eta_T \geq 0$. Consider the one-dimensional function $f : \mathbb{R} \to \mathbb{R}$ given by $f(x) = \cos(x)$. The function $f$ is both 1-Lipschitz and 1-smooth, and since $f$ is bounded between $-1$ and $1$, any initial suboptimality is bounded by 2. The update at step $t$ can be written as

$$x_{t+1} = x_t - \eta_t f'(x_t) = x_t + \eta_t \sin(x_t).$$

Let $H_t(x) = x + \eta_t \sin(x)$. Since $H_t$ is continuous and $|H_t(x) - x| \leq \eta_t$, we have $H_t(x) \geq x - \eta_t \to \infty$ as $x \to \infty$, and $H_t(x) \leq x + \eta_t \to -\infty$ as $x \to -\infty$. Hence, by the intermediate value theorem, for every $y \in \mathbb{R}$ there exists $x \in \mathbb{R}$ such that $H_t(x) = y$.

We construct an initialization for which the last iterate is not stationary. Set $x_{T+1} = \pi/2$, and define the preceding iterates recursively by choosing any solution to

$$H_t(x_t) = x_{t+1}, \qquad t = T, T-1, \ldots, 1.$$

Such a choice is possible by the previous surjectivity observation. By construction, running gradient descent from this $x_1$ produces exactly the sequence $x_1, \ldots, x_{T+1}$, and in particular its last iterate is $x_{T+1} = \pi/2$. However,

$$|\nabla f(x_{T+1})| = |-\sin(\pi/2)| = 1.$$

Thus, for every horizon $T$ and every prescribed stepsize sequence, there exists an initialization such that the last iterate of gradient descent is not even approximately stationary. Since deterministic gradient descent is a special case of SGD, this rules out a general last-iterate stationarity guarantee for SGD in the non-convex smooth setting without further assumptions.

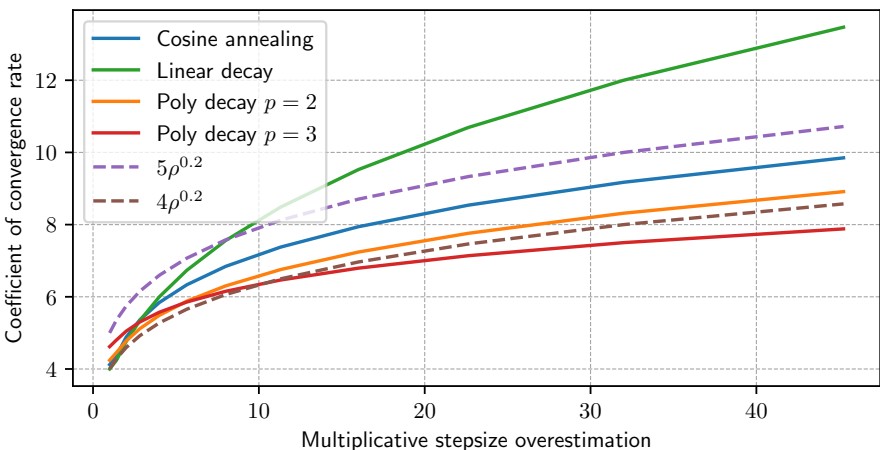

*Figure 3.* Numerically evaluating the coefficient of $DG/\sqrt{T}$ for the convergence guarantee of Theorem 1 with different schedules and varying multiplicative misspecification parameter $\rho$.

## F. Tighter constants using numerical analysis

The constants of Corollary 3 are not tight; in particular, the bound is established using crude (up to constants) bounds for $H_h(u)$ and $Q_h(u)$. While a tighter bound can be obtained, the framework easily yields to numerical analysis as we demonstrate next.

The convergence guarantee of Theorem 1 is not posed as a closed-form equation but rather as a minimization over integrals that depend on the schedule. For a specific schedule and misspecification parameter, we use Scipy's (Virtanen et al., 2020) quad integration to evaluate $H_h, Q_h$, and fsolve to solve the minimization of Theorem 1.

In Figure 3 we provide a numerical analysis for the convergence guarantee of Theorem 1 with several decaying schedules, including the cosine annealing, showing in particular that the convergence rate of SGD with cosine annealing is bounded by $5\rho^{\frac{1}{5}}\frac{DG}{\sqrt{T}}$. We observe that the cosine annealing schedule and the quadratic decay schedule have similar convergence guarantees with a coefficient between $4\rho^{\frac{1}{5}}$ to $5\rho^{\frac{1}{5}}$. In addition, even for a somewhat large misspecification parameter of size 50, the difference between cosine annealing and the different polynomial decaying schedules is at most a factor of 2, which indicates that even mild decay might be sufficient if the grid is not too coarse.

## G. Proofs of Section 3

### G.1. Proof of Corollary 3

Note that $h(u)$ is non-increasing, differentiable ($h'(u) = \frac{-\pi}{2}\sin(\pi u)$), $\frac{\pi}{2}$-Lipschitz (as $|h'(u)| \le \frac{\pi}{2}$) and satisfy $h(u) = 0 \Leftrightarrow u = 1$. Hence, $h$ is *annealed* and by Theorem 1,

$$\mathbb{E}[f(x_{T+1}) - f(x^\star)] \le \frac{1}{2}\mathsf{Rate}_{h,T}^{\mathsf{tu}} \cdot \inf_{\tau \in [0,1)}\left(\frac{H_h(0)}{\rho H_h(\tau)} + \frac{\rho Q_h(\tau)}{Q_h(0)}\right) + O\left(\frac{\rho \eta_{\mathsf{tu}}G^2}{T}\right).$$

Next, we will bound $h(u), H_h(u)$ and $Q_h(u)$ using polynomials. As $\cos(\pi - \theta) = -\cos(\theta)$ and $\cos(\theta) \ge 1 - \frac{\theta^2}{2}$,

$$h(u) = \frac{1}{2}(1 - \cos(\pi(1 - u))) \le \frac{\pi^2}{4}(1 - u)^2 \le \frac{5}{2}(1 - u)^2. \tag{14}$$

On the other hand, for $u \in [0, 1)$,

$$\left(\frac{h(u)}{(1 - u)^2}\right)' = \frac{-\frac{\pi}{2}\sin(\pi u)(1 - u)^2 + 2(1 - u)}{(1 - u)^4} = \frac{4 - \pi(1 - u)\sin(\pi u)}{2(1 - u)^3} \ge \frac{4 - \pi}{2} > 0.$$

Using the fundamental theorem of calculus, for all $u \in [0, 1)$,

$$\frac{h(u)}{(1-u)^2} = \frac{h(0)}{(1-0)^2} + \int_0^u \left(\frac{h(v)}{(1-v)^2}\right)' dv \geq 1 + \int_0^u 0 \cdot dv = 1 \implies h(u) \geq (1-u)^2. \tag{15}$$

Using integration, Equations (14) and (15) also implies that

$$\frac{1}{3}(1-u)^3 \leq H_h(u) \leq \frac{5}{6}(1-u)^3. \tag{16}$$

Using the above inequalities,

$$Q_h(v) = \int_v^1 \frac{h(u)^2}{H_h(u)} du \leq \frac{75}{4} \int_v^1 (1-u) du = \frac{75}{8}(1-v)^2 \tag{17}$$

and

$$Q_h(v) \geq \int_v^1 \frac{6(1-u)^4}{5(1-u)^3} du = \frac{3}{5}(1-v)^2. \tag{18}$$

Using the bounds, setting $\bar{\tau} = 1 - \rho^{-0.4} \in [0, 1)$, and noting that $H_h(0) = \frac{1}{2}\int_0^1 (1 + \cos(\pi u)) du = \frac{1}{2}$,

$$\frac{H_h(0)}{\rho H_h(\bar{\tau})} + \frac{\rho Q_h(\bar{\tau})}{Q_h(0)} \leq \frac{3}{2\rho(1-\bar{\tau})^3} + \frac{\rho 125(1-\bar{\tau})^2}{8} = \frac{3}{2\rho\rho^{-1.2}} + \frac{125\rho\rho^{-0.8}}{8 Q_h(0)} \leq 18\rho^{0.2}. \tag{19}$$

Thus,

$$\mathbb{E}[f(x_{T+1}) - f(x^\star)] \leq \mathsf{Rate}_{h,T}^{\mathsf{tu}} \cdot 18\rho^{\frac{1}{5}} + O\left(\frac{\rho \eta_{\mathsf{tu}} G^2}{T}\right).$$

Again noting that $H_h(0) = \frac{1}{2}$ and using Equation (17),

$$\mathsf{Rate}_{h,T}^{\mathsf{tu}} = \frac{2DG}{\sqrt{T}}\sqrt{Q_h(0)/H_h(0)} = \frac{2DG}{\sqrt{T}}\sqrt{2Q_h(0)} \leq \frac{2DG\sqrt{\frac{150}{8}}}{\sqrt{T}} \leq \frac{10DG}{\sqrt{T}}. \qquad \square$$

