# OpenReview forum: "Learning Rate Annealing Improves Tuning Robustness in Stochastic Optimization"
_ICML.cc/2026/Conference — ICML 2026 regular_

### Official Review · Reviewer_byor · 2026-03-06

**Soundness:** 3
**Presentation:** 3
**Significance:** 3
**Originality:** 3
**Overall Recommendation:** 5
**Confidence:** 3

**Summary:**

This work explored a new perspective on the benefits of annealing schedules, revealing their improved robustness to initial stepsize overestimation. Theoretical analysis under the stochastic convex optimization setting demonstrates that the (last-iterate) convergence rate of stochastic gradient descent with annealed schedules depends sublinearly on the multiplicative misspecification factor $\rho$, achieving a rate of $O(\rho^{1/(2p+1)}/\sqrt{T})$ where $p$ is the degree of polynomial decay and $T$ is the number of steps. Experiments confirm the increased robustness compared to tuning with a fixed stepsize, suggesting that annealing schedules are more robust to coarse grid searches.

**Compliance With Llm Reviewing Policy:**

Affirmed.

**Final Justification:**

The counterexample for fixed step-size SGD in the non-convex case provided in the rebuttal is simple yet clear. I therefore raise my rating and remain positive about this paper.

**Key Questions For Authors:**

The results of this work appear interesting. As mentioned in the weaknesses section, is it possible to extend the current theoretical analysis to the non-convex Lipschitz or non-convex smooth setting? Although it may be difficult to establish last-iterate convergence guarantees in the non-convex setting, such an extension could further broaden the applicability of the results. The major challenge I believe is that the techniques used in Appendix C may not apply to the non-convex setting.

**Limitations:**

Yes

**Strengths And Weaknesses:**

**Strengths**

- This work is technically sound, clearly written and well structured.

- A novel term $\text{Rate}_{h,T}^{\text{tu}}$ is introduced to unify the theoretical analysis and the derived convergence guarantee is for the last iterate of SGD. A lower bound is also established to show that linear degradation in misspecification factor $\rho$ is unavoidable for fixed stepsize SGD.

- Experiments on synthetic logistic regression task with a linear model and CIFAR-10 classification task with a deep neural network validate annealing schemes (such as cosine annealing and linear decay) demonstrate greater robustness compared to a fixed step size schedule.

**Weaknesses**

- This work mainly focuses on the convex and Lipschitz setting, as well as the convex and smooth setting. While the theoretical analysis and the newly introduced metrics (such as $\text{Rate}_{h,T}^{\text{tu}}$) are clean and novel, the current results do not extend to the non-convex setting, whereas modern large-scale models and neural networks are typically non-convex.

- The experiments section does not include the schedule-free (SGD) baseline introduced by [1].

[1] Defazio, A., Yang, X., Mehta, H., Mishchenko, K., Khaled, A. and Cutkosky, A., 2024. The road less scheduled. Advances in Neural Information Processing Systems, 37, pp.9974-10007.

---

> ### Author Rebuttal · Authors · 2026-03-31
>
> Thank you for your review. Please see below our responses to the main points you raised.
>
> > “This work mainly focuses on the convex and Lipschitz setting ... the current results do not extend to the non-convex setting, whereas modern large-scale models and neural networks are typically non-convex.”
>
> While we agree that the non-convex case is indeed relevant to the loss landscape of neural networks, it is also a very general, unstructured setting where strong, meaningful convergence statements are hard to come by. This is one reason why convex Lipschitz (stochastic) optimization is a primary focus of theoretical research in optimization/ML, and serves as a canonical testbed of optimization methods and phenomena. There are a multitude of theoretical research papers published every year (in NeurIPS/ICML/…) in this area. Our paper belongs to this literature, with the main goal of theoretically studying an empirical observed phenomenon within a fundamental, well-studied optimization framework.
>
> > “The experiments section does not include the schedule-free (SGD) baseline introduced by [1].”
>
> Our goal in this paper is studying the (tuning-robustness) effect of various step size schedules on vanilla SGD, so as to better understand it theoretically. From this perspective, a comparison with a different, schedule-free method (such as the one in [1], which replaces the schedule by performing queries at convex combinations of points, and thus not based on the vanilla SGD template) seems to be mostly orthogonal to this study. But in case we misunderstood your intention, we’d be happy to be corrected during the discussion.
>
> > “... is it possible to extend the current theoretical analysis to the non-convex Lipschitz or non-convex smooth setting?”
>
> This is a good question. Unfortunately, in the non-convex case, it is generally impossible to obtain a last-iterate guarantee for SGD, on which our results strongly rely. The following is a brief proof sketch of a counterexample for fixed step size SGD (we will include a formal proof in the revision if this seems of interest):
>
> Let $f(x)=\cos x$ and consider GD with stepsize $\eta \in (0,1)$, for which $x_{t+1}=x_t+\eta\sin x_t$. Let $h(x)=x+\eta\sin x$. Thus, $h'(x)=1+\eta\cos x \ge 1-\eta>0$, implying that $h$ is strictly increasing and hence invertible.
>
> Fix any horizon $T$ and set $x_T=\pi/2$. Let $x_t =h^{-1}(x_{t+1})$ for $t=T-1,\dots,1$. By construction, the last iterate of GD, starting at $x_1$, is $x_T=\pi/2$. Hence, the last iterate satisfies $|\nabla f(x_T)|=|-\sin(\pi/2)|=1$.
>
> Thus, for every $T$, there exists an initialization such that $x_T=\pi/2$, where $|\nabla f(x_T)|=1$, and therefore $x_T$ is not a stationary point.

---

> > ### Author Rebuttal · Reviewer_byor · 2026-04-02
> >
> > I thank the authors for the rebuttal. The counterexample for fixed step-size SGD in the non-convex case is simple but clear. I have raised my rating accordingly.

---

> > > ### Author Response · Authors · 2026-04-04
> > >
> > > We sincerely thank you for your feedback and your positive assessment of our work. We will include the counterexample in our final version.

---

### Official Review · Reviewer_Jten · 2026-03-12

**Soundness:** 2
**Presentation:** 2
**Significance:** 3
**Originality:** 4
**Overall Recommendation:** 4
**Confidence:** 4

**Summary:**

The paper explores advantages of annealed learning rate schedules, focusing on their robustness to initial stepsize overestimation. The paper assumes convex optimization solved by gradient descent. The mainly theoretically driven findings suggest that learning rate annealing behaves more robustly compared to grid searches in this sense. The authors provide a two empirical justification on a toy example, and CIFAR-10 training using a WideResNet.

**Compliance With Llm Reviewing Policy:**

Affirmed.

**Final Justification:**

I agree that studying optimization in convex Lipschitz settings is well-motivated. However, I remain cautious about how well insights from such simplified settings translate to the practical scenarios, which I consider most important. I have adjusted my score accordingly. That said, I do not oppose acceptance of this paper.

**Key Questions For Authors:**

1. My major concern is the applicability of the idea to broader sense. Currently, the paper is mainly focusing on a narrow scope of theory and practice. Generalizing the results to the real world practices may seem to be trivial to the authors, but I believe this should be justified in the paper to get the full credit. This is because, the theory is based on Lipschitz continuous setting with convex objects, which corresponds to a very small proportion in the real world applications.
2. How about different type of schedules rather than cosine annealing? We have exponential decay, linear decay, polynomial decay, and many others. Although these options are implied in lines 175-189, we still need an extensive empirical studies that justifies the generalizability of the results.
3. Another significant alternative is to do the LR annealing multiple times from maximum to the minimum, overcoming the local minima. How can the suggested results explain or fit to this well-used practice?

**Limitations:**

Yes.

**Strengths And Weaknesses:**

### Strengths

1. (Originality) The paper justifying the LR annealing schedules through robustness perspective opens a new avenue of investigate for the better optimization.
2. (Significance) Relating the robustness criteria of LR scheduling/annealing with the sensitivity to step size misspecification is a nice, worth-explorable area of research.
3. (Presentation) The paper delivers rich theoretical justification to the raised questions and insights regarding robustness to initial stepsize overestimation.

---

### Weaknesses & Suggestion

1. (Soundness) The theoretical findings are based on convex settings, only. This can be an easy surrogate assumptions that allows us to explore in this field, but this oversimplification may overlook the initial motivation of these annealing practices. That is, to overcome local minima, not appearing in the convex settings.
2. (Presentation) Since the theory tackles a minimal, sounding scenarios, the experiments should be extensive to support the authors claim. However, having only a WideResNet experiment on CIFAR-10 does not seem to fully deliver the practical value of this work.
3. (Presentation) The authors may put the proofs into the appendices, and instead focus on more real world experiments and empirical analysis. Currently, the paper’s main manuscript can be extended to better support the claim through richer experiments.

---

> ### Author Rebuttal · Authors · 2026-03-31
>
> Thank you for your review. Please see below our responses to the main points you raised.
>
> > “The theoretical findings are based on convex settings ... allows us to explore in this field ... this oversimplification may overlook the initial motivation of these annealing practices ... overcome local minima ...”
>
> There are several motivations for LR schedules in general, and LR annealing in particular. A prominent modern interpretation of annealing is as a replacement to model (suffix) averaging [1,2], and our theoretical study stems from this perspective. This viewpoint is unrelated to avoiding local minima and remains relevant in the convex case.
>
> That said, we agree that there are other aspects of LR schedules beyond iterate averaging, last-iterate guarantees and tuning robustness, but we cannot cover all practical implications of complex learning pipelines.
>
> > “Since the theory tackles a minimal, sounding scenarios, the experiments should be extensive to support the authors claim ...”
>
> The primary contribution of our work lies in optimization theory, and we view its theoretical significance as standing on its own. The small-scale experiments we include are intended to provide some validation of the new theory we propose, rather than to offer comprehensive empirical evidence. There is already ample empirical evidence suggesting annealed LR schedules work well in practice, and the goal of our work is to shed some light **theoretically** on why this is the case.
>
> > “The authors may put the proofs into the appendices, and instead focus on more real world experiments and empirical analysis. Currently, the paper’s main manuscript can be extended to better support the claim through richer experiments.”
>
> We respectfully disagree. This is primarily a theoretical paper, and we consider the proofs to be an integral, principal part of the main body, serving to present our theoretical ideas and contributions.
>
> > “My major concern is the applicability of the idea to broader sense ... narrow scope of theory and practice ... the theory is based on Lipschitz continuous setting with convex objects, which corresponds to a very small proportion in the real world applications.”
>
> We have to respectfully disagree on this account as well. Convex Lipschitz (stochastic) optimization is a primary focus of theoretical research in optimization/ML, and serves as a canonical testbed of optimization methods and phenomena. There are a multitude of theoretical research papers published every year (in NeurIPS/ICML/…) in this area. Our paper belongs to this literature, and again - with the main goal of theoretically studying an empirical observed phenomenon within a fundamental, well-studied optimization framework.
>
> > “... Generalizing the results to the real world practices may seem to be trivial to the authors ...”
>
> On the contrary! By no means do we intend to belittle the difficulties of real-world practice, which pose many challenges that our theory, and optimization theory at large, has yet to address. This is, by the way, a main reason for studying this kind of questions in a controlled theoretical model like Lipschitz continuous convex optimization, so as to isolate particular challenge/phenomenon we wish to understand better.
>
> > “How about different type of schedules rather than cosine annealing? We have exponential decay, linear decay, polynomial decay ... Although these options are implied in lines 175-189, we still need an extensive empirical studies that justifies the generalizability of the results.”
>
> Our theoretical results consider the fixed schedule, annealing schedules (including cosine annealing, linear decay, and the general family $(1-t/T)^p$, where $t$ is the step, $T$ is the horizon, and $p\geq 1$ is a constant), as well as the $1/\sqrt{t}$ schedule. Our empirical results, however, cover the commonly used subset of fixed, linear decay, and cosine annealing.
>
> The focus of this work is to introduce a new theoretical framework and enriched understanding (primarily of existing practices), and it does not aim to provide a comprehensive empirical evaluation of all schedules.
>
> > “Another significant alternative is to do the LR annealing multiple times from maximum to the minimum, overcoming the local minima. How can the suggested results explain or fit to this well-used practice?”
>
> Our theoretical framework supports the widely popular use of cosine annealing (as well as linear decay). That said, there are other aspects of learning rate schedules that are not captured by our tuning-robustness perspective, including learning rate schedules with restarts and/or how annealing can be used to avoid convergence to local minima.
>
> [1] Jain, P., Nagaraj, D., and Netrapalli, P. Making the last iterate of sgd information theoretically optimal. COLT19
>
> [2] Ge, Rong, Sham M. Kakade, Rahul Kidambi, and Praneeth Netrapalli. "The step decay schedule: A near optimal, geometrically decaying learning rate procedure for least squares." NeurIPS19

---

> > ### Author Rebuttal · Reviewer_Jten · 2026-04-03
> >
> > Thank you for the rebuttal. I acknowledge the authors' standpoint on the value of theoretical study in convex Lipschitz (stochastic) optimization settings. I will raise my score accordingly.

---

> > > ### Author Response · Authors · 2026-04-04
> > >
> > > Thank you for your thoughtful feedback and your positive assessment of our work, particularly sharing the view of LR scheduling as a means of escaping local minima.

---

### Official Review · Reviewer_WXiq · 2026-03-13

**Soundness:** 3
**Presentation:** 3
**Significance:** 3
**Originality:** 3
**Overall Recommendation:** 4
**Confidence:** 3

**Summary:**

This paper interprets the advantage of learning rate annealing from the perspective of hyperparameter robustness. It shows that without LR annealing, increasing the LR beyond the optimal LR by a ratio of $\rho$ causes performance decay linear in $\rho$, but with LR annealing, the performance decay becomes sublinear.

**Compliance With Llm Reviewing Policy:**

Affirmed.

**Final Justification:**

After the rebuttal, I believe that the paper is well grounded and well written. The theoretical part decouples the benefits of LR scheduling from model averaging, which is a significant contribution. The theoretical parts are also verified by experiments. Although there can be concerns about the narrow scope, I think the authors are doing well enough in exploiting this direction, and the insights overweigh the drawbacks. I would therefore recommend acceptance.

**Key Questions For Authors:**

1. The experiments use the Nesterov momentum while the theoretical analysis does not consider momentum. How will momentum possibly influence the theoretical result
2. The experiments show that LR decay provides advantage different from averaging. Can this be justified by theory?
3. [1] argues that large LR actually helps to accelerate the convergence of logistic regression. However, the analysis of [1] depends on the assumption that the optimal model parameters are sufficiently bounded away from initialization. Does it indicate that the theory of this submission breaks when $D\to\infty$?

[1] Wu et al. Large Stepsize Gradient Descent for Logistic Loss: Non-Monotonicity of the Loss Improves Optimization Efficiency. 2024.

**Limitations:**

Yes.

**Strengths And Weaknesses:**

## Strengths

1. The paper is well written with carefully discussed assumptions and theorems.
2. The interpretation from hyperparameter robustness is intersting.

## Weaknesses

1. Some more intuitions of the function $Q$ is expected.
2. The theoretical results are not **quantitatively** verified in experiments, as is confessed in Section 4.3. Specifically, the polynomial decay with $p=2$ (so that $\rho^{2p+1}=\rho^{1/5}$) should be similar to cosine schedule, and empirical verification of this connection would be very exciting. I will raise my score if the theoretical results are better justified quantitatively. I feel fine if the authors find the test loss less robust and present the training loss instead.
3. The main setting of this paper involves increasing the learning rate of SGD beyond the optimal LR. This motivation is a bit confusing even with Footnote 5, which argues the motivation from the overshooting effect of grid search. Suppose that the grid search tests learning rates like $\eta_k=2^k\cdot \eta_0$, then the effect of overshooting is in the order of a constant.

---

> ### Author Rebuttal · Authors · 2026-03-31
>
> Thank you for your review. Please see below our responses to the main points you raised.
>
> > “The main setting of this paper involves increasing the learning rate of SGD beyond the optimal LR. This motivation is a bit confusing .. tests learning rates like $\eta_k=2^k \cdot \eta_0$ .. order of a constant.”
>
> This seems to be a misunderstanding (that we will definitely clarify further in the revision). The reviewer suggested one extreme, where $\eta_k =2^k\cdot\eta_0$ and overshooting occurs only by a small constant factor, in which case tuning robustness has a limited effect. On the other hand, for very large models, performing even a much coarser grid search is computationally prohibitive, and scaling laws are used instead, where the learning rate may be arbitrarily larger than the optimal one.
>
> In a middle ground, one may have a budget to use $\eta_k =c^k\cdot\eta_0$ for some $c$ that can be much larger than 2,10, or even 50. This corresponds to a large multiplicative factor, and having a sublinear dependence on it can be highly beneficial.
>
> > “The experiments show that LR decay provides advantage different from averaging .. justified by theory?”
>
> Yes, our theory suggests **exactly that**: while both LR annealing and averaging offer an optimal $O(1/\sqrt{T})$ convergence rate, only LR annealing provides improved tuning robustness. This is precisely one of our primary contributions, and is **quantitatively verified in our experiments**. As we mention in the paper (e.g., lines 76 and 104, second column) and show in Appendix D, averaging alone is not sufficient for achieving a sub-linear dependence on a multiplicative overestimation. We would be happy to further discuss this point if additional details are needed.
>
> > “.. polynomial decay with $p=2$ .. similar to cosine schedule .. I will raise my score if the theoretical results are better justified quantitatively ..”
>
> Thank you for bringing this up. We originally omitted the experiments with the quadratic schedule as it behaves similarly to cosine annealing (though when carefully tuned it performs just slightly worse), and we agree that the similarities are interesting to discuss. We conjecture that the initial phase of cosine annealing, in which the decay is slower than quadratic decay (1/2 at t=T/2 rather than 1/4) is beneficial for performance, even though they decay similarly later on.
>
> Follow [this link](https://anonymous.4open.science/r/anon37875544-ADB4) for matching figures to Fig 1. (right) and Fig 2. (right), with additional plot for the quadratic decay schedule. We will include these in the revision, of course - thanks again.)
>
> > “Some more intuitions of the function $Q$ is expected.”
>
> Thank you for the suggestion; we will include more intuition about the $Q$ function. In a nutshell: given a schedule $h$ and assuming $\int_u^1h(v)dv\propto (1-u)h(u)$ (which holds for the fixed schedule and cosine annealing), we have $Q_h(v) \propto \int_v^1(h(u)/(1-u))du$. In this case, a good schedule (i.e., one with a small $Q$) is one that becomes very small toward the end of training, so that for $1-u \ll 1$, $h(u)/(1-u)$ is small. This is precisely the behavior of annealing schedules, whereas the fixed schedule diverges. We would be happy to further clarify in the discussion period if needed.
>
> > “The experiments use the Nesterov momentum .. theoretical analysis does not consider momentum .. influence the theoretical result”
>
> The synthetic experiments do not use momentum and are more closely aligned with the existing theory. Regarding the CIFAR-10 experiments, which involve non-convex problems and are therefore already outside the strict theoretical setting, we note that gaps between theory and practice are common, and this is one such instance, as we follow a standard training recipe known to perform well on this task.
>
> We conjecture that similar theoretical results hold with fixed momentum; however, we are not aware of general last-iterate guarantees for SGD with fixed momentum. Extending our results (and the last-iterate guarantees) requires further research.
>
> > “[1] .. large LR actually .. accelerate .. logistic regression .. theory of this submission breaks when \$D \to inf$?”
>
> While the setting of [1] (which even allows for accelerated rates) is quite different from the convex-Lipschitz setting we consider, your observation about $D \to \infty$ is correct. As we explicitly mentioned in the limitations section (paragraph starting at line 419, second column), our bounds depend on $D$ rather than $\lVert x_1-x^\star\rVert$. Note that having a known (and relatively tight) diameter bound is a common limitation of quite a few adaptive and “parameter-free” methods in the literature [2,3].
>
> [2] Ali Kavis, Kfir Y Levy, Francis Bach, and Volkan Cevher. UniXGrad: A universal, adaptive algorithm with optimal guarantees for constrained optimization. NeurIPS19
>
> [3] R. Ward, X. Wu, and L. Bottou. Adagrad stepsizes: Sharp convergence over nonconvex landscapes. ICML19

---

> > ### Author Rebuttal · Reviewer_WXiq · 2026-04-02
> >
> > Many thanks for the rebuttal. My concerns are fully resolved. Special thanks for the quantitative experiments. I have increased my rating.

---

> > > ### Author Response · Authors · 2026-04-04
> > >
> > > Thank you for your response.
> > >
> > > We appreciate your helpful comments and your positive assessment of our work, and will address your suggestions in the final revision.

---

### Decision · Program_Chairs · 2026-04-30

**Decision:**

Accept (regular)

**Comment:**

This paper provides a clear theoretical perspective on learning-rate annealing, interpreting its benefit through robustness to stepsize overestimation. Reviewers found the theoretical contribution technically strong, particularly the sublinear robustness guarantees under annealing and the separation from fixed-step behavior.

The rebuttal strengthened the paper. The authors added the quantitative experiments requested by Reviewer **WXiq** and clarified several scope and theory questions, after which Reviewers **WXiq**, **Jten**, and **byor** all stated that their concerns were resolved and raised or maintained positive scores.

While the work is intentionally limited to a controlled convex setting, it does not undermine its theoretical contribution. Overall, I find the paper worthy of acceptance.